

# Chemical precursors of new particle formation in coastal New Zealand

Maija Peltola[1], Clémence Rose[1], Jonathan V. Trueblood[1], Sally Gray[2], Mike Harvey[2], and Karine Sellegri[1]

[1]Laboratoire de Météorologie Physique (LaMP-UMR 6016, CNRS, Université Clermont Auvergne), 63178, Aubière, France
[2]National Institute of Water & Atmospheric Research Ltd (NIWA) 301 Evans Bay Parade, Greta Point, Wellington New Zealand

**Correspondence:** Maija Peltola (m.peltola@opgc.fr) and Karine Sellegri (k.sellegri@opgc.univ-bpclermont.fr)

**Abstract.** To reduce uncertainties in climate predictions, we need to better understand aerosol formation in different environments. An important part of this is studying which chemical species are responsible for particle formation. While many advances have been made in this field, measurements are lacking especially from marine environments. Here, we measured the chemical composition of ambient ions over 7 months at Baring Head station, located in coastal New Zealand. This adds to our

previous work which reported the aerosol size distribution measurements and investigated new particle formation and environmental conditions favouring new particle formation at the station. By combining the information on ion chemical composition with our previous work, we were able to study the chemical precursors of new particle formation. Our results showed that while over land new particle formation is likely driven by sulfuric acid and organic species, in clean marine air iodine oxides and sulfur species are likely important drivers of particle formation processes. These data were also used to characterise the

diurnal and seasonal cycles of most important ion groups and their geographical source regions. Sulfate ions displayed a clear daytime maximum where as iodine oxides had morning and evening maximums. Highly oxygenated organic molecules on the other hand, were most abundant during the night when the air was land-influenced. This data set is highly valuable and our results provide important information on the chemical species driving new particle formation at a remote Southern Hemisphere coastal site.

## 1 Introduction

Understanding aerosol formation is important for accurate climate predictions, because aerosols and their cloud interactions form some of the largest uncertainties in predicting future climate (Stocker et al., 2013) and new particle formation has been shown to form approximately half of the cloud condensation nuclei (CCN) globally (Gordon et al., 2017). Despite the fact

that oceans cover over 70% of the Earth, most aerosol measurements are from continental sites in the Northern Hemisphere (e.g. Kerminen et al., 2018). Marine aerosol sources include sea spray formation from breaking waves and secondary aerosol



formation from chemical species emitted from the sea surface. Several recent publications have pointed to secondary marine aerosols being more important for the climate than sea spray in favourable conditions (Quinn et al., 2017; Sanchez et al., 2018; Fossum et al., 2020).

The potential importance of secondary marine aerosol formation in regulating climate was first pointed out by Charlson et al. (1987), who proposed that dimethyl sulfide (DMS) produced by phytoplankton could form new particles after being oxidised to sulfuric acid, thus influencing clouds and climate. Since then, many studies have examined the emissions, reaction products and aerosol yields of DMS. The most commonly considered oxidation products of DMS are methane sulfonic acid (MSA) and $SO_2$, with $SO_2$ further oxidising into sulfuric acid (SA). The relative yields of these compounds vary depending on

conditions such as temperature, $NO_X$ levels and halogen concentrations (Nicovich et al., 2006; Stark et al., 2007; Breider et al., 2010; Mardyukov and Schreiner, 2018) and the uptake of MSA to aerosol phase can depend on the chemical composition of particles (Yan et al., 2020). One factor complicating understanding of DMS oxidation processes is multiphase chemistry. Some work has showed that aqueous phase formation of MSA can be more important than its gas phase formation (Hoffmann et al., 2016; Baccarini et al., 2021). Recent research has also found new DMS oxidation products, such as hydroperoxy thioformate

(HPMTF, $HOOCH_2SCHO$, Veres et al., 2020) and methane sulfonamide (MSAM, $CH_5NO_2S$, Edtbauer et al., 2020). Veres et al. (2020) suggested that HPTMF could participate in aerosol formation, but some more recent work has indicated that the formation of HPMTF would rather be a sink of sulfur and reduce the formation of secondary aerosol rather than increase it (Wollesen de Jonge et al., 2021; Novak et al., 2021). This shows that more information is needed on the importance of different DMS oxidation products in the real atmosphere.

Many studies in the past decades have shown that DMS is not the only species that is responsible for marine aerosol formation. For example, iodine was shown to form new particles already 20 years ago (O'Dowd et al., 2002). Advances have been made since then to understand the mechanisms of iodine aerosol formation in detail (Sipilä et al., 2016; Gómez Martín et al., 2020; He et al., 2021). Gómez Martín et al. (2020) showed that the initial steps of particle formation are driven by bimolecular and third-body reactions of $I_xO_y$ and He et al. (2021) showed that particle formation can be driven by $HIO_3$, while

$HIO_2$ is important for stabilising $HIO_3$ clusters during the first steps of nucleation. Iodine can be emitted to the atmosphere in different forms as a result of both biological processes and sea surface chemistry (Carpenter et al., 2021). While iodine oxides (Takashima et al., 2022) and iodine in aerosols (Gómez Martín et al., 2021) have been observed globally, iodine emissions are specifically high from macroalgae at some coastal sites (Carpenter et al., 2000; McFiggans et al., 2004; Sellegri et al., 2005; McFiggans et al., 2010). This is why much of the work studying iodine particle formation has focused on the effect of

coastal sources of iodine (Grose et al., 2007; Furneaux et al., 2010; Sipilä et al., 2016). However, iodine particle formation has been observed also over sea ice (Baccarini et al., 2020) and indications of it happening over open ocean (Sellegri et al., 2016) and other environments (He et al., 2021) exist as well. Despite the recent progress in understanding aerosol formation from iodine, real atmospheric conditions are complicated and field measurements are needed to understand the importance of these processes and the interactions between iodine species and other chemical species. For example, recent modelling work

suggested that MSA could stabilise iodic acid, leading to higher particle formation rates when both MSA and iodic acid are present (Ning et al., 2021), but this has not yet been reported in measurements. Field work in coastal China has shown that





there nucleation can involve both iodine and organic species (Wan et al., 2020), further highlighting the need to understand the interactions between different chemical species.

Other species shown to be able to participate in marine new particle formation include ammonia (Jokinen et al., 2018), amines (Brean et al., 2021), and several organic species. Ammonia and amines are both known to stabilise sulfuric acid (Yao et al., 2018; Lehtipalo et al., 2018) and MSA (Chen and Finlayson-Pitts, 2017). Organic species have been suggested to play a significant role in the marine CCN budget as well (Mayer et al., 2020; Zheng et al., 2020). Organic species that could be important for marine particle formation include for example monoterpenes ($C_{10}H_{16}$), isoprene ($C_5H_8$), and different organic acids. Monoterpenes are known to be an important aerosol precursor species due to their capability to form highly oxygenated organic molecules (HOM, see e.g., Bianchi et al., 2019) and they have been observed in marine aerosols at least by Cui et al. (2019). Isoprene was first suggested to be a source of marine secondary organic aerosol (SOA) by Meskhidze and Nenes (2006) and it has been further studied by, for example, Arnold et al. (2009) and Cui et al. (2019). While the emissions of monoterpenes and isoprene are typically connected directly to biological activities (Shaw et al., 2010), isoprene can be produced also abiotically by photosensitized reactions at the sea surface microlayer (Ciuraru et al., 2015). Isoprene oxidation can lead to formation of dicarbonyls glyoxal and methylglyoxal which could play a role in SOA formation (Lawson et al., 2015). Reactions at the sea surface can also produce other volatile organic compounds that can form aerosols, such as unsaturated aldehydes (Ciuraru et al., 2015; Rossignol et al., 2016) and various other species (Brüggemann et al., 2017).

This study provides information on chemical precursors of aerosols at Baring Head site in coastal New Zealand. Measurements of chemical composition of aerosols or their precursors in this region are rare. Most of the previous work at the station has focused on filter samples (Allen et al., 1997; Sievering et al., 2004; Li et al., 2018, 2021), and they have highlighted the role of sulfate formation. A ship campaign east of New Zealand also indicated that formation of SOA could be an important aerosol formation pathway during plankton blooms (Law et al., 2017). The closest long-term aerosol measurements are from Cape Grim, Australia. There, secondary aerosol has been shown to consist of secondary sulfate and nitrates (Crawford et al., 2017) with a small contribution from isoprene- and monoterpene-derived SOA (Cui et al., 2019). We report the chemical composition of ambient ions in coastal New Zealand over a 7 month period. We combine these data with the results from our previous study that focused on new particle formation at the site and showed that NPF can occur in open ocean air masses (Peltola et al., 2021) to understand which chemical species drive new particle formation in both marine and land-influenced air. This is the first time that the chemical composition of ambient ions has been measured in the area and our results provide valuable information for understanding the air chemistry and NPF precursors at this site and in marine air in general.

## 2 Methods

### 2.1 Measurements

To understand which chemical species drive new particle formation in coastal New Zealand, we performed extensive measurements of aerosol properties and chemical composition of ambient ions at the Baring Head station which is located on the south coast of New Zealand's North Island. Baring Head is known for its long term measurements of $CO_2$ and the station was chosen





for this purpose because it regularly receives air masses coming from the Southern Ocean that have not been in touch with land for several days (see e.g., Stephens et al., 2013). The ability to sample open ocean air masses is why we chose the station for our study.

This article follows the work of Peltola et al. (2021) and most of the measurements are described in detail by them. Briefly, the aerosol measurements included measurements of aerosol particle size distribution between 10-500 nm with a Scanning

Mobility Particle Sizer (SMPS), ion and particle size distribution measurements with a Neutral cluster and Air Ion Spectrometer (NAIS, Mirme and Mirme, 2013) and aerosol particle number concentration measurements for particles with diameters above 10 nm with a Condensation Particle Counter (CPC) and for particles larger than 1 and 3 nm with a Particle Size Magnifier (PSM, Vanhanen et al., 2011). Additionally, we used the long term measurements of meteorological variables available from the NIWA climate data base (https://cliflo.niwa.co.nz/, last accessed May 2021) and radon, tide height, and air mass back

trajectory data as described by Peltola et al. (2021).

In addition to the measurements described by Peltola et al. (2021), we measured the chemical composition of ambient ions with an Atmospheric Pressure Interface Time of Flight mass spectrometer (APi-TOF, Junninen et al., 2010). We used the instrument in negative ion mode without chemical ionisation. The benefits of using an APi-TOF without chemical ionisation include the fact that the signal is not influenced by the choice of the ionising chemical species and that the detection limit is

several orders of magnitude lower than with a chemical ionisation inlet (see e.g., Beck et al., 2022). The APi-TOF was located in the same building with the aerosol instrumentation and it was used with a 91 cm long 3/8" thick stainless steel inlet. The flow rate to the instrument was increased to 11 lpm with an external pump to reduce wall losses in the inlet. The measurements were done with 1 min time resolution, but here we use data averaged over 30 min to increase the signal to noise ratio. The time period with usable data from the APi-TOF extended from 16 July 2020 to 17 January 2021 and in total we had 117 d 12 h of

usable APi-TOF data. The monthly distribution of available APi-TOF data is summarised in Table 1.

**Table 1.** Percentage of APi-TOF data available for each month calculated based on 30 minute averaged data.

| Month | July 2020 | August 2020 | September 2020 | October2020 | November 2020 | December 2020 | January 2021 |
|---|---|---|---|---|---|---|---|
| (%) | 50 | 88 | 68 | 98 | 45 | 10 | 26 |

## 2.2 Data analysis

As in our previous work, all the data were divided into marine and land-influenced air based on air-mass back trajectories and wind and radon data. In short, marine air consists of air masses that have not, based on the air-mass back trajectories, crossed over land in the past three days, have wind directions between 120–220° and radon values below 100 mBq m$^{-3}$. The rest of the

data was considered land-influenced. While in our previous work 7.3 % out of the aerosol data were classified as clean marine air (Peltola et al., 2021), out of the APi-TOF data only 4.3% classified as clean marine air because of lower data availability.



Air-mass back trajectory data were also used to determine potential source areas of different chemical species following what has been done before with aerosol data by Rose et al. (2015) and Peltola et al. (2021). Briefly, the idea is to give each air mass back trajectory the value that the signal of a given chemical species has at the time of arrival of the air mass at the site and average the values in each map grid cell so that if air masses arriving from a certain region have higher signals than others, this shows on the map. Only cells with at least 10 back trajectories passing through them were used.

The aerosol data were used to calculate aerosol particle and ion number concentrations in different size ranges, to classify the days based on if new particle formation was observed, and to calculate aerosol growth rates. Growth rates were determined for nucleation (< 25 nm), Aitken (25-100 nm) and accumulation (100-500 nm) mode particles with the algorithm by Paasonen et al. (2018). For new particle formation event analysis, we used the criteria by Dal Maso et al. (2005). A more detailed description of the aerosol data analysis and its results can be found in Peltola et al. (2021).

The APi-TOF data were processed with Tofware software (https://www.tofwerk.com/software/tofware/, last accessed October 2021). We used Tofware to fit all peaks that had reasonable signals and then identified the most relevant compounds with correlation analysis. For this we used so-called 'schemaball' plots (https://se.mathworks.com/matlabcentral/fileexchange/42279-okomarov-schemaball, last accessed June 2021) that illustrate the correlations between all the peaks and aerosol properties. In these plots, each line corresponds to a correlation between two different variables and the colour of the line is relative to the strength of the correlation (See Fig. 2). Only correlation coefficients with p-values smaller than 0.05 are drawn. The idea of these plots is to visualise the interactions between all the selected peaks and to create useful compound groups for further analysis.

# 3 Results and discussion

## 3.1 Detected species

Figure 1 shows an example mass spectrum of APi-TOF data summed over 24 h. The largest peaks consist of nitrate ion ($NO_3^-$), its water cluster, bisulfate ($HSO_4^-$) and its cluster with sulfuric acid. This is similar to what has been observed previously with an APi-TOF in a boreal forest (Junninen et al., 2010). Pure sulfuric acid clusters were detected up to the trimer, which is what has been observed earlier with later similar measurements in a boreal forest as well (Yan et al., 2018), but contrary to their work, we saw no sulfuric acid clustered with ammonia.

In addition to the highest peaks, we can observe clear peaks for chloride, iodate and different organic compounds. Around m/Z 300–400 Th we can see peaks that correspond to highly oxygenated organic molecules (HOM) and are likely monoterpene ($C_{10}H_{16}$) oxidation products (see e.g., Yan et al., 2016). The highest peaks in this mass range corresponded to $C_{10}H_{14}O_7NO_3^-$, $C_{10}H_{14}O_9NO_3^-$, $C_{10}H_{16}O_9NO_3^-$, and $C_{10}H_{14}O_{11}NO_3^-$, which have earlier been observed to be the highest peaks in a boreal forest and in $\alpha$-pinene oxidation chamber experiments as well (Ehn et al., 2012). The majority of the observed peaks were below m/Z 400 Th with barely any peaks observed above m/Z 500 Th. The lack of peaks at higher masses can be explained by a combination of clean air, losses in the inlet and potential non-optimal tuning of the instrument. Baccarini et al. (2021) had similar issues of not detecting expected peaks at higher masses in the Southern Ocean close to Antarctica despite using



chemical ionisation. Future studies in clean air should aim to further lower the detection limits of the instrument or to better

optimise instrument settings so that compounds with larger masses could be observed and identified as well.

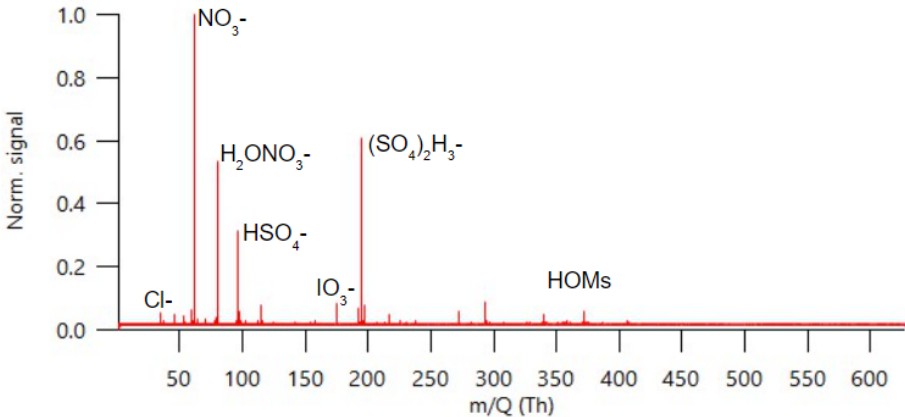

**Figure 1.** Example of APi-TOF spectrum summed over 24 h with the signal normalised to one.

### 3.1.1 Identifying relevant compounds

In addition to identifying major peaks in the mass spectrum, we fitted all the peaks that we could find in the APi-TOF data

(see Table A1 for full peak list) and grouped potentially interesting species based on their correlations with each other and

aerosol data. To facilitate this analysis, we used schemaball plots which illustrate the correlations between all the peaks (see

full explanation in Section 2.2). For these plots we used all the peaks in APi-TOF data and particle number concentrations (N)

and growth rates (GR) in different size ranges to see if some compounds correlate with these variables. This was done because

later we want to connect the chemical species to aerosol formation.

We drew these plots separately for both marine and land-influenced air and for the whole day and different times of the

day (0-5 and 8-15 h NZST, not shown here). Figure 2 shows the schemaball for all data in land-influenced air. Many of the

observed negative correlations can be explained by the diurnal variations of the compounds, as will be further discussed in

Section 3.2. For example, $NO_3$- has a negative correlation with bisulfate because it is abundant during the night while sulfuric

acid is produced during the day. Positive correlations can be seen for compounds with similar chemical species, such as $IO_3$-

and $H_2IO_4$, compounds with sulfuric acid, and different HOMs. In marine air (Fig. 3), the observed negative correlations are

primarily similar to land-influenced air. For positive correlations, the most striking difference to land-influenced data is the lack

of strong correlations between different HOMs, likely resulting from lower signals of HOMs in marine air (See Figs. 4 and 5).

Based on these two figures and similar figures limited to night- (0-5 h) and daytime (8-15 h) data, we defined 11 groups

of identified compounds and 4 groups of unidentified ions for further analysis. These groups are summarised in Table 2. The

groups with known compounds include groups for the most abundant ions, nitrates ($\Sigma NO_3$) and sulfuric acid ($\Sigma SA$), species

known to be important for chemistry in marine air, such as methane sulfonic acid (MSA), different halogens, and different



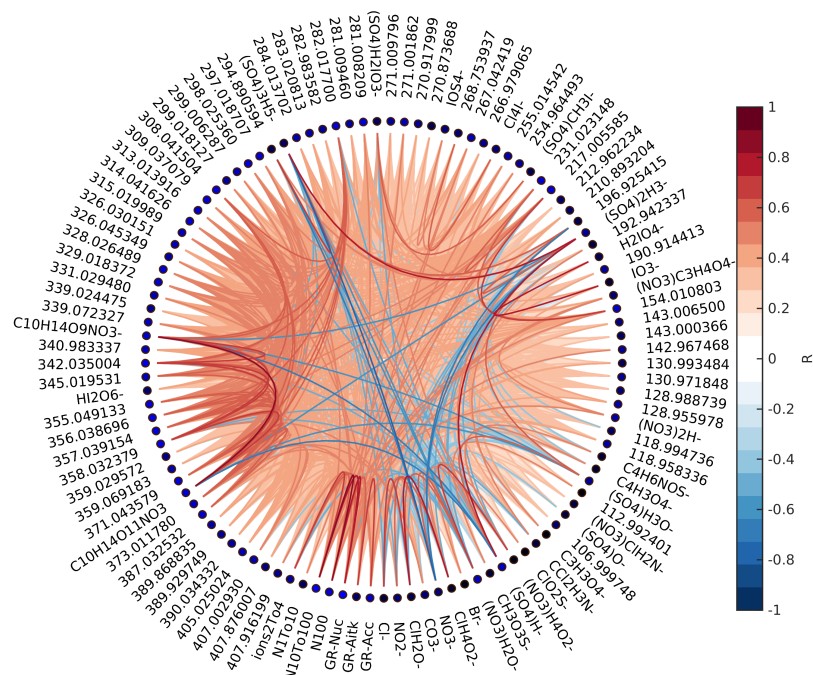

**Figure 2.** Spearman's correlation coefficients (R) between different APi-TOF peaks, particle number concentrations and growth rates in different size ranges in land-influenced air. Each line corresponds to one correlation coefficient and the strength of the correlation defines the colour of the line. Lines are drawn only for correlation coefficients with p-values below 0.05. The colour of the node of each compound (blue or black dots next to the compound label) is determined based on the average absolute correlation so that compounds with on average higher correlations have brighter colours.

organic groups. For iodine oxides, there are two groups because the compounds in the first group ($\Sigma$ I) correlated positively with each other while $HI_2O_6$- was not observed as often as the two other iodine species and out of the species that we identified, it was the iodine oxide with the highest mass. For organic species there are three groups ($\Sigma$ HOM, $\Sigma$ Malonate, and Maleate) since the sources of different organic species could be different. Malonic acid is a saturated dicarboxylic acid and maleic acid

is an unsaturated dicarboxylic acid. While malonic acid has been found to be the second most abundant dicarboxylic acid in the Southern and Pacific Oceans, concentrations of maleic and fumaric acids are typically an order of magnitude lower (Wang et al., 2006). Dicarboxylic acids can be derived from isoprene emissions and unsaturated fatty acids at the sea surface (Bikkina et al., 2014), which can both be related to new particle formation (Alpert et al., 2017; Meskhidze and Nenes, 2006). It is also well known that HOMs are important for aerosol formation (Bianchi et al., 2019). This is why all of these compounds are of

interest and relevant for studying new particle formation.

The 4 groups of unidentified ions were chosen so that we would not miss potentially important compounds. Peaks in group "Other1" showed strong correlations in marine air. "Other2" contains peaks that had relatively strong correlations with each

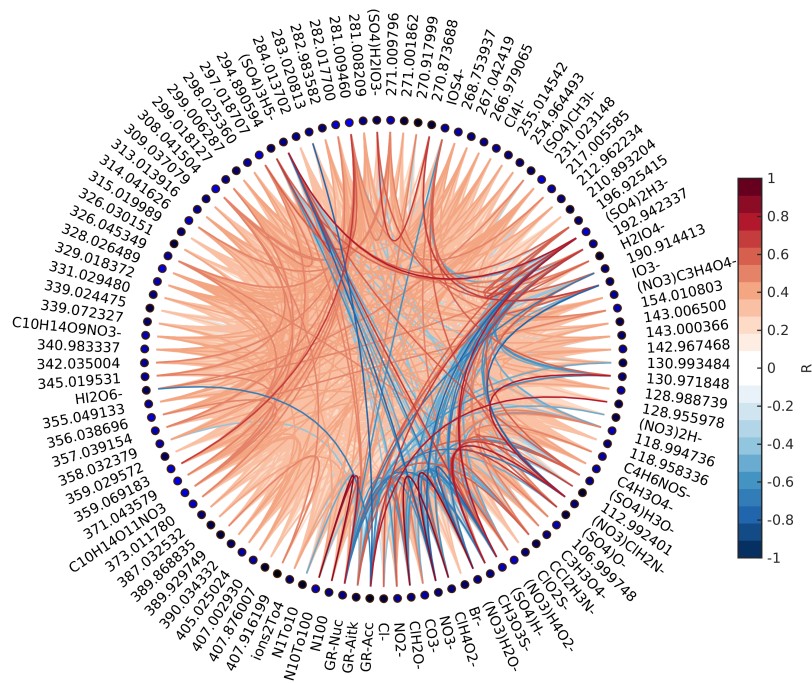

**Figure 3.** Spearman's correlation coefficients (R) between different APi-TOF peaks, particle number concentrations and growth rates in marine air. See Figure 2 for explanation.

other and aerosol concentrations in marine air during the night. "Other3" contains peaks that showed correlations in land-influenced air. "Other4" contains peaks that had a positive correlation with nucleation mode growth rates in land-influenced

air during the day. At least some of these peaks are likely organics, since they are in the same mass range with the identified HOMs and the peaks did not match for example any iodine oxide or sulfuric acid peaks that we would be aware of.

### 3.1.2   General comparison of marine and land-influenced air masses

To gain some understanding on where different compounds come from, Figure 4 illustrates how the average signals of selected ions differ in marine and land-influenced air masses by showing the ratio of average signals of marine ion groups compared to

the average signal of the same groups in land-influenced air. Most of the ions are more abundant in marine air, which highlights the importance of ion losses to aerosol population in land-influenced air. The most significant difference can be observed for MSA, which has around 13 times higher signal in marine air. This is not surprising since MSA comes from marine sources and when the air has spent time over land, MSA from the ocean has likely had time to condense on the surface of aerosols from terrestrial sources. Chloride and bromide are also relatively high in marine air which is expected since they have marine

sources as well.





**Table 2.** Compounds group used to analyse the data.

| Group name | Peaks included |
| --- | --- |
| $\Sigma\ NO_3$ | $NO_3^-$, $NO_3H_2O^-$, $(NO_3)_2H^-$ |
| $\Sigma\ SA$ | $HSO_4^-$, $(SO_4)_2H_3^-$, $(SO_4)_3H_5^-$ |
| MSA | $CH_3O_3S^-$ |
| $\Sigma\ Cl$ | $Cl^-$, $ClH_2O^-$, $ClH_4O_2^-$ |
| $Br^-$ | $Br^-$ |
| $\Sigma\ I$ | $IO_3^-$ and $H_2IO_4^-$ |
| $HI_2O_6^-$ | $HI_2O_6^-$ |
| $\Sigma\ S+I$ | $(SO_4)CH_3I^-$, $(SO_4)H_2IO_3^-$ |
| $\Sigma\ HOM$ | $C_{10}H_{14}O_7NO_3^-$, $C_{10}H_{14}O_9NO_3^-$, $C_{10}H_{16}O_9NO_3^-$, $C_{10}H_{14}O_{11}NO_3^-$ |
| $\Sigma\ Malonate$ | $C_3H_3O_4^-$, $(NO_3)C_3H_4O_4^-$ |
| Maleate | $C_4H_3O_4^-$ (Maleate(1-) or fumarate(1-)) |
| Other1 | m/Z 281.009, 359.069 |
| Other2 | m/Z 143.006, 271.009, 299.006, 326.045 |
| Other3 | m/Z 154.011, 217.006, 231.023 |
| Other4 | m/Z 331.029, 339.024, 345.020, 373.012 |

The only species that seem to be higher in land-influenced air are organics and unknown compounds, which are also likely organics. The difference is clearest for HOMs and considering that the masses we observed here are similar to those observed in a boreal forest (e.g., Sulo et al., 2021), their likely source is forests. The only organic group in this analysis that has higher signals in marine air is maleate. It can be produced by photochemistry and although it also has anthropogenic precursors, it

could be derived from compounds produced in biologically active marine areas (Sempéré and Kawamura, 1996), which would explain why it is more abundant in marine air.

## 3.2 Diurnal cycles

To understand the processes controlling the signals of key ion species, we plotted the diurnal cycles of the ion groups separately for clean marine and land-influenced air (Fig. 5). The important thing to remember when looking at ion signals is that they

are driven not only by changes of concentrations of neutral species, but also by charge availability and losses. In practice this means that, for example, sulfuric acid is relatively likely to be charged due to its low proton affinity and it can 'steal' charges





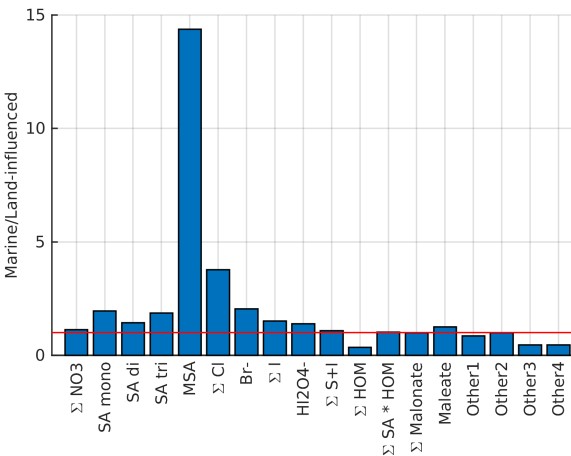

**Figure 4.** Comparison of average ion signals in marine and land influenced air masses. Red horizontal line indicates 1:1 relationship.

from other chemical species, leading to the diurnal cycles of ions being different from those of neutral species (see e.g., Ehn et al., 2010; Bianchi et al., 2017). Here, we will focus on other factors that can be important for the diurnal cycles. In this section it should be also remembered that we have less data in marine air compared to land-influenced air, meaning that the
diurnal cycles in marine air can be more uncertain.

The first examined ion group is the sum of nitrates (Fig. 5a). Nitrate is one of the most common ions in the atmosphere (Eisele, 1989). The nitrate ions have a clear diurnal cycle with lower daytime signals and increase during the night. This is because during the day, $NO_3$ is destroyed by photolysis (e.g., Wayne et al., 1991). The diurnal cycles of nitrates are similar in marine and land-influenced air. During the day, around 11-13 LT, marine air has slightly higher signals than land-influenced
air. A possible explanation for this is that total ion losses in marine air are lower than in land-influenced air, because the total aerosol surface area is on average lower (see, Peltola et al., 2021).

The second ion group is the sum of bisulfate ions and bisulfate ions clustered with sulfuric acid ($\Sigma$ SA, Fig. 5b). Bisulfate ions are also one of the most abundant ambient ions observed. For $\Sigma$ SA, we observe a clear diurnal cycle in both land-influenced and clean marine air masses. The signals increase in the morning after 5 am, reach a maximum around midday,
and decrease in the evening. This is a very typical cycle for sulfuric acid because its production in unpolluted environments is driven by the oxidation of $SO_2$ and other sulfur species by OH radicals in the presence of sunlight, and its losses are driven by condensation to aerosol particles and other surfaces (see e.g., Tanner and Eisele, 1991; Dada et al., 2020). If we compare the diurnal cycles of SA in marine and land-influenced air masses closely, we can see that SA increases in both air masses around the same time, but in marine air, the median SA signals reach higher values in the afternoon and decline later in the evening
than in land-influenced air. This can be explained by smaller losses to aerosol surface and thus longer SA lifetimes in clean marine air. This is in line with results observed earlier when comparing observations at Mauna Loa, in Hawaii and Cheeka Peak Research Station, in western coastal USA (Tanner and Eisele, 1991).

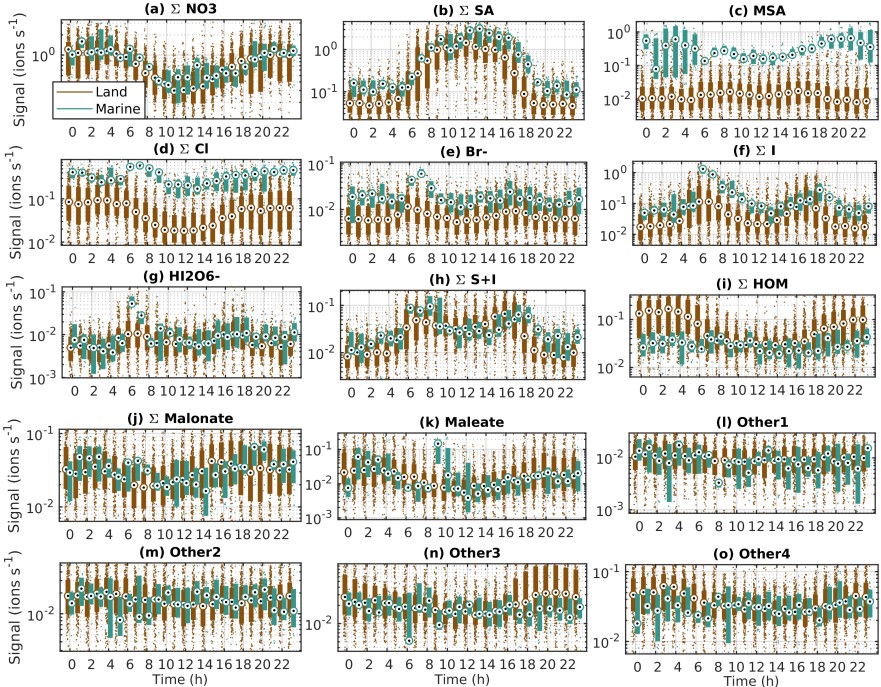

**Figure 5.** Diurnal cycles of example compounds at Baring Head in clean marine and land-influenced air. The circles are the median signals for each hour, boxes mark 25th and 75th percentiles and the rest of the points are outside this range.

Another major sulfur compound is methane sulfonic acid (MSA, Fig. 5c). MSA is clearly higher in marine air and displays only a weak diurnal cycle in both air masses. In marine air, MSA levels are variable and reach the highest values during the
night whereas in land-influenced air the signal varies less. Possible explanations for the marine diurnal cycle include daytime production masked by charge loss to sulfuric acid (Tanner and Eisele, 1991) and MSA being produced in aqueous phase and later released to gas phase (Baccarini et al., 2021; Berresheim et al., 2002).

The next ions are halogen species. For the chloride group we can see that the signals are clearly higher in marine air and that in both air mass classes the signals are lower during the day (Fig. 5d). The source of chloride is likely sea spray and
the surface of the ocean, which explains the higher signals in marine air. For bromide, the marine signals are again higher than land-influenced air signals (Fig. 5e) because of the marine sources of halogens. In both air masses we can see morning and evening peaks, with the marine morning peak being the most distinguishable. The morning increase indicates that more bromide is formed when photochemistry is triggered. The midday decrease could be related to for example charge loss to sulfuric acid. Similar results have been seen before at Jungfraujoch, a high altitude station in Europe (Frege et al., 2017).

For the most abundant iodine oxides ($\Sigma$ I, Fig. 5f) we can observe two peaks in both marine and land-influenced air masses, as was seen for bromide. The iodine oxide signals are in general higher than bromide signals and the diurnal cycle is even clearer. The first peak occurs around 7 h NZST and the second around 17 h NZST. This is somewhat similar to the patterns





observed for IO$_3$- at Jungfraujoch by Frege et al. (2017). In marine air masses the peaks are typically higher than in the land-influenced air, especially for the morning peak. This is reasonable since the sources of iodine oxides are likely marine and the ion sink is higher over land. As for sulfuric acid, we can see that the decrease of iodine oxide peaks is slower in marine air, which is likely explained by the longer lifetimes due to smaller sink to particle surfaces.

The fact that we observe peaks in the morning and evening is likely explained by photochemistry producing these iodine oxides during the day. One possible explanation for the daytime decrease is photolysis or reactions with other chemical species such as halogen atoms or OH· radicals (Frege et al., 2017). Another possible explanation is again the charge stealing by sulfuric acid. While SA formation requires UV light (Eisele and Tanner, 1990, 1993) and its formation can be disturbed even by passing clouds, iodine oxides can form even in cloudy conditions (He et al., 2021). This could explain why iodine oxides start forming earlier than SA. On the other hand, Baccarini et al. (2021) observed similar morning and evening peaks for iodic acid with nitrate chemical ionisation in the Southern Ocean. They hypothesised that this would be due to less iodate produced when the radiation levels are higher. This is supported by results by Sellegri et al. (2016) who observed bi-diurnal iodine peaks in aerosol chemical composition data from mesocosm experiments.

For the only identified iodine oxide with a higher molecular weight, HI$_2$O$_6$-, the signals are lower than for the lighter iodine oxides and the diurnal cycle is less clear (Fig. 5g). We can, however, still see the highest signals of this compound in marine air around the same time as for the lighter iodine oxides. This shows that if the conditions are favourable, larger iodine oxide molecules can be present at detectable quantities and follow similar patterns to the smaller ones. For compounds containing both sulfur and iodine species, we can see that the signal is lowest during the night and the highest peaks occur roughly at the same time with iodine oxide peaks (Fig. 5h).

Out of the organic compounds examined, the clearest diurnal cycle can be observed for the $\Sigma$ HOM group (Fig. 5i) in land-influenced air. The signals of these HOMs are highest during the night. This is reasonable since all the compounds in this group were charged by nitrate ions and nitrates are more abundant during the night. This diurnal cycle is similar to the diurnal cycle of similar ions in a boreal forest (Bianchi et al., 2017). HOMs could also be charged by sulfate ions during the day (see e.g., Bianchi et al., 2017), but we did not find any peaks corresponding to organosulfates. Apart from chemical processes, one factor contributing to higher nighttime HOM signals can be the height of the planetary boundary layer. During the night, the boundary layer is typically lower than during the day, meaning that the emissions from ground level are mixed to a smaller volume of air.

The other identified organic compounds were the dicarboxylic acids, malonate and maleate (Figs. 5j and k). They both have low signals and the levels are similar in marine and land-influenced air masses. In land-influenced air, the signals have a morning minimum and then increase over the day. Previous work has shown that malonic acid can be produced by photochemical oxidation (Kawamura and Sakaguchi, 1999), so the increase during the day seems reasonable. In marine air, we can also observe peaks during the morning around 7-8 h and evening around 19-20 h for malonate and one peak around 9-10 h for maleate, but otherwise the cycle is similar to land-influenced air. The times of the peaks of malonate are similar to the iodine oxide peaks in marine air. The fact that highest hourly median values of malonate and maleate are observed in clean marine





air is not surprising since malonic acid has been connected to marine sources previously (Kerminen et al., 2000; Röhrl and Lammel, 2001).

Out of the 'Other' compound groups (Figs. 5 l-o), the first two have low signals and do not show clear diurnal variations
or clear differences between land-influenced and marine air masses. Other3 increases in the evening in land-influenced air and then decays overnight. This is somewhat similar to the land-influenced maleate signal although the evening maximum seems stronger for this group. The Other4 group also shows some higher nighttime signals in land-influenced air, but this increase is observed later, with highest signals observed only after midnight. This is similar to what was seen for the HOM group and the masses in this group are also in a similar mass range with the HOM group, supporting the assumption that these compounds
are likely unidentified HOMs.

## 3.3 Seasonal cycles

In this section, we study the seasonal cycles of different chemical compounds to expand our understanding of the sources of different chemical species. We use only 10 groups as opposed to the 15 groups in the previous section as groups with similar compounds (e.g. iodine oxides) are likely to have similar seasonal cycles. Even though we lack data for part of the year and
especially for the summer months the data are sparse, seasonal cycles can be observed for some of the compounds (Fig. 6).

The first compound group in Figure 6 is nitrate ions. They show some month-to-month variations, but no clear seasonal cycle. This is not surprising since nitrate is always abundant in the atmosphere. Sulfuric acid, on the other hand, is higher during October-December in land-influenced air (Fig. 6b). In January, the signal is lower, but our data availability for this month is limited. The increase towards summer can be explained by the increase in radiation levels and higher DMS emissions
from the ocean due to higher biological activity in the area (see e.g., Li et al., 2018). In marine air, we cannot see a similar trend. One reason for this could be the fact that we have a lot less data from the clean marine periods. Another possibility is that when SA production is higher, more aerosols are formed, which increases the losses of sulfuric acid. In marine air masses, the changes in the losses can be more significant than in land-influenced air, where the losses are practically always higher than in marine air. Our previous work at Baring Head showed that sub-100 nm particles have a clear seasonal cycle in marine air
with a maximum in the summer, whereas in land-influenced air the difference between spring and summer is less clear (Peltola et al., 2021).

MSA increases towards the summer and has highest median values during the spring in both air mass classes (Fig. 6c). This can be explained by marine biogenic sources and availability of radiation. As the sources of MSA are marine, we can assume that the MSA in land-influenced air is from marine sources as well. The spring maximum is in line with previous work at
Baring Head and the Chatham Rise area, which has shown increased sulfate production during the late spring and summer (Li et al., 2018; Allen et al., 1997; Law et al., 2017).

Chloride seems to increase towards the summer in both marine and land-influenced air masses, but the trend is small compared to variations within single months (Fig. 6d). The reason for this trend is not clear, but it could be related to photochemical release of chlorine from the sea surface or sea spray aerosol surfaces and chloride displacement from reaction of sea-salt aerosol



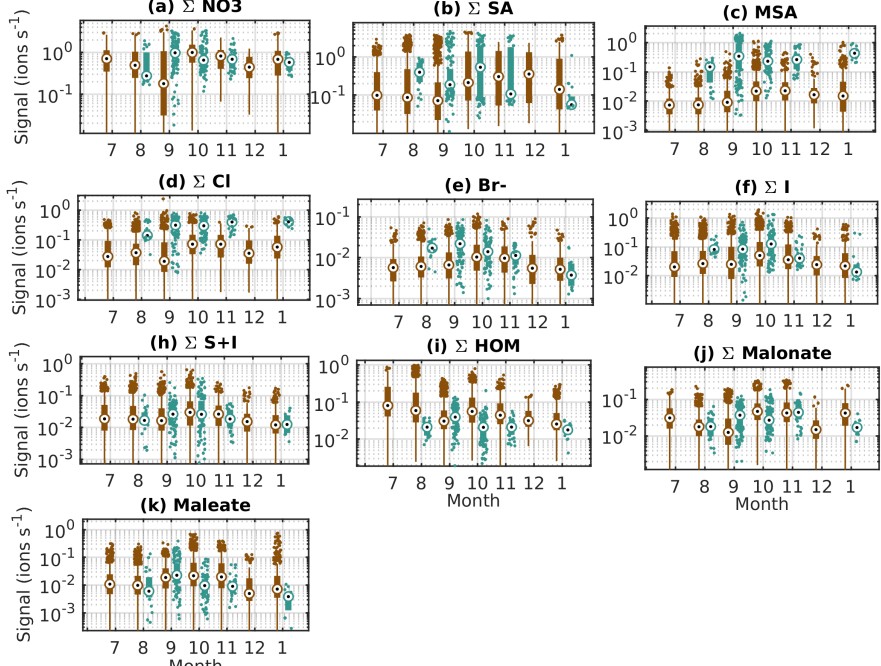

**Figure 6.** Seasonal cycles of example compounds at Baring Head in clean marine and land-influenced air. The circles are the median signals for each month, black boxes mark 25th and 75th percentiles, the whiskers mark 1.5 times the interquartile range and rest of the points are outside this range.

with more abundant summertime sulfuric and nitric acids (Wang et al., 2019). With bromide we observe a spring maximum in land-influenced air and a decrease from spring to summer in marine air (Fig. 6e).

Iodine oxide levels are highest in both air mass classes during the spring (Fig. 6f). The trends are similar but weaker for compounds containing both sulfur and iodine (Fig. 6h). Baccarini et al. (2021) suggested that iodic acid concentrations should be higher in the spring and autumn compared to summer if its formation is favoured by lower light levels. Our results would

support this interpretation. Another possible factor increasing the levels of iodine species during the spring could be increased biological activity in the ocean.

The HOMs are highest during the winter in land-influenced air and have no clear seasonal patterns in marine air (Fig. 6i). The winter maximum is interesting considering that monoterpene emissions are typically higher at higher temperatures (Guenther et al., 1993). One potential reason for the higher wintertime signals could be lower boundary layer heights during

the winter. Also, the HOMs that we have in this group were all charged by nitrates and most abundant during the nighttime and in the winter the nights are longer. The highest malonate signals in marine air are observed in the spring (Fig. 6j), which points to a marine biological source. In land-influenced air, malonate signals are more variable, but the highest median levels are observed during the spring. Maleate signals are slightly higher in the spring in both marine and land-influenced air masses which would point to biological sources as well (Fig. 6k). Our previous work showed that while new particle formation can be





observed at the site all year round, the highest fraction of non-events was observed during the winter and sub-100 nm particle number concentrations were higher during the spring and summer. If we compare that information to the seasonal cycles of different chemical species, at least sulfur and iodine species are known to participate in particle formation and have higher signals during the spring and summer, indicating that they could be potential particle precursor species at Baring Head. This is further examined in the following sections.

## 3.4   Geographical source regions


To find out if different geographical regions were responsible for emitting more particle producing vapours (e.g. if biologically active marine areas are responsible for some species), we drew source maps for the most relevant ions with the method described briefly in Section 2.2. Here, it should be noted that even though we examine only grid cells with at least 10 air-mass back trajectories passing through them, the grid cells on the edges of the shown area likely include less data than the ones in

the middle, and might in turn be more affected by the observations associated to a single day. Here, we use all air masses rather than limiting the data to only marine air, because limiting the air masses to only clean marine air would leave us with too little data.

    The first map is for sulfuric acid dimer (Fig. 7a). We chose to display the dimer rather than the sum of all peaks or the monomer, since the monomer is typically the highest peak and is always abundant, meaning that little geographical differences

can be seen with it. The dimer has also been identified as a good indicator of NPF in a more polluted environment (Cai et al., 2021), making it more interesting for our analysis. The highest SA dimer signals are observed northwest of New Zealand. One possible reason for this is the transport of $SO_2$ from Australia. Some of the patches with higher SA signals could also be related to shipping, which has previously been identified as the second most important source of non-sea-salt sulfate aerosol at Baring Head with DMS oxidation being the most important source (Li et al., 2018). One likely significant source of shipping

emissions is the Wellington harbour, which is approximately 15 km northwest from the station.

    MSA is also a DMS oxidation product, but unlike SA, it does not have anthropogenic sources. With MSA, we can see clear regional differences (Fig. 7b). Highest MSA signals are observed when air masses come from the direction of the Southern Ocean. One possible explanation for this is transport of DMS from the Southern Ocean and Antarctic coast. For example, Baccarini et al. (2021) observed higher concentrations of MSA at latitudes higher than 60°and link this to higher production

of DMS close to Antarctica and lower temperatures increasing yield of MSA over sulfuric acid. Here, it should be noted that if MSA or its precursors are originated from close to the Antarctic coast, MSA would have needed to travel to Baring Head in aqueous or particle phase and evaporate closer to the station since the lifetime of gaseous MSA is estimated to be less than an hour (Baccarini et al., 2021). Another possible reason for higher MSA signals when the air masses come from south is that on their way to the station they pass over the biologically active Chatham Rise area which can produce DMS (Law et al., 2017, see

e.g.,). One more possible explanation is also higher losses of MSA when the air comes from other directions. The air-masses coming from the south have typically not been in contact with land unlike the air-masses arriving from other directions and if the air mass passes over land, the particle concentrations and thus MSA sink increase.

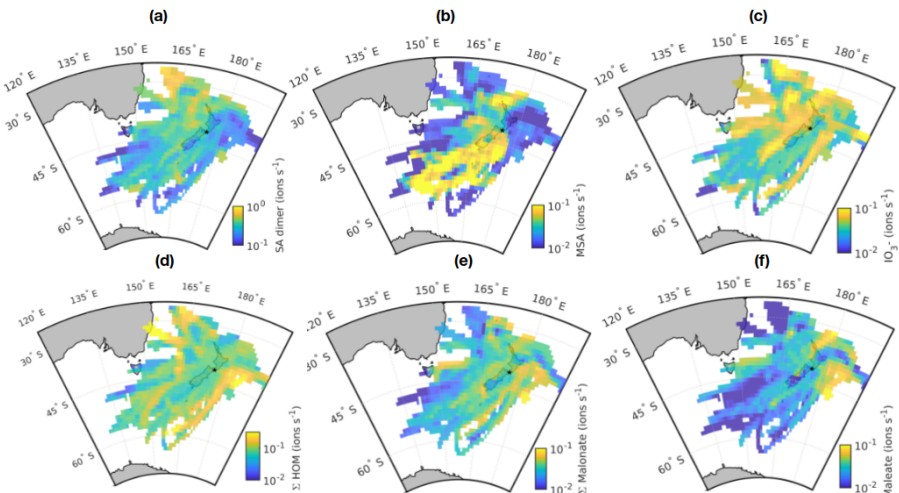

**Figure 7.** Source areas for a) sulfuric acid dimer, b) methanesulfonate (MSA), c) IO$_3$-, d) $\Sigma$ HOM, e) malonate and f) maleate. The station is marked with a small black star. See text for full description.

For iodate, the patterns are less straightforward to explain. Figure 7c shows that there are stripes of higher iodate signals in several different directions. One possible explanation could be coastal sources of iodine and thus increased iodate signals

when the air masses pass over certain regions. While iodine emissions can be high in some coastal environments such as Mace Head (e.g. Dall'Osto et al. (2011)), iodine emissions depend on both algae types (e.g., Carpenter et al., 2000) and how much the macroalgae are exposed to air when tides change. For example, previous work at Cape Grim has shown that there iodine emissions from the local macroalgae are significantly smaller than at Mace Head and that the emissions only have a rather local effect on air chemistry (Grose et al., 2007; Cainey et al., 2007). In our data, the correlation between tide height and iodate

signal during the day seems very small if not negligible (R = -0.037 p = 0.038), meaning that the main source of iodine is likely open ocean rather than coastal sources.

Figure 7d shows the source area plot for HOMs. One of the areas with the highest signals is coming from the direction of Australia, meaning that either monoterpenes get transported all the way from Australia or that the air masses coming from this direction pass a monoterpene source for example at the southern tip of North Island. Two other areas with higher HOM signals

appear east and west of New Zealand. The area to the east is partly similar to the higher iodate signal area. The reason for this is not clear, but part of these trajectories could correspond to air masses that come from the ocean but pass over the North Island during the last day before reaching the station as is shown later in the case study for land-influenced NPF. Both HOMs and iodate could also have biologically active marine sources.

Figure 7e shows the source map for malonate. The signals of malonate are similar in all directions but seem highest when

the air masses come from the east side of New Zealand or pass close to the west coast of New Zealand. As discussed earlier, dicarboxylic acids can be produced by photochemistry and biologically productive marine areas are one of their possible





sources, which is possible here as well. With maleate, we see higher signals east of the station and north of North Island (Fig. 7f). This could be related to biologically active marine areas or air masses crossing over the North Island.

### 3.5 Chemical precursor species of new particle formation

To understand which chemical species are responsible for particle formation at Baring Head, we combine the information on the chemical composition of ions with aerosol data. First, we go through some case studies and compare the average chemical composition of ions on days with and without new particle formation. Finally, we explore the correlations between different chemical species and aerosol data. This is done separately for land-influenced and marine air.

#### 3.5.1 Land-influenced new particle formation

To start the analysis, we look into an example of typical new particle formation day at Baring Head. We use 10 October 2020 as an example, since this was a clear new particle formation event based on the criteria by Dal Maso et al. (2005). The particle size distribution for this day is shown in Figure 8a. The 72 h air-mass back trajectories show that the air mass originated from the ocean south of New Zealand, likely from the free troposphere, but later passed over the North Island (Fig. 8b) and the particle formation event that we observe likely occurred during the time that the air mass spent over land. At the beginning

of the day we can see an Aitken mode that was formed by NPF during the previous day. Around 11 h NZST we see a new nucleation mode appearing and growing past 25 nm by early afternoon and continuing to grow after 20 h.

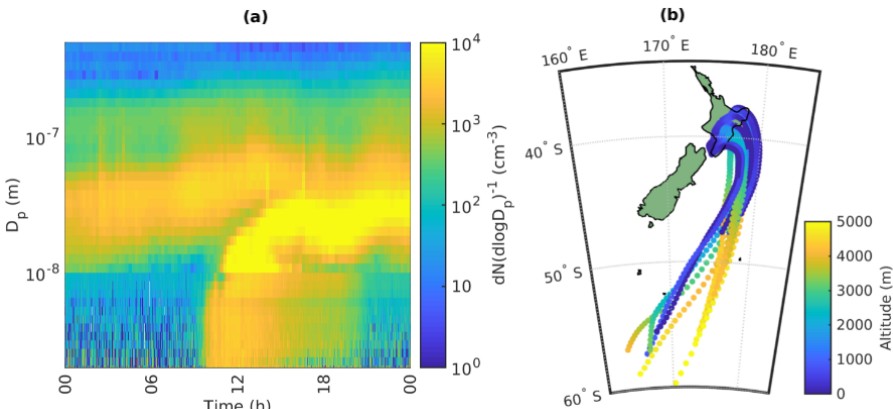

**Figure 8.** a) Particle number size distribution with combined NAIS (below 10 nm) and SMPS (10-500 nm) data during October 10th 2020. b) Air-mass back trajectories coloured by altitude for the same day.

Figure 9 shows the time evolution of different ions on the same day. Here, we use the same groups that we used for the diurnal cycles before, but separate the sum of sulfates to bisulfate ion (SA mono), bisulfate ion clustered with one sulfuric acid (SA di) and bisulfate ion clustered with two sulfuric acid molecules (SA tri), because the dimer is a better indicator for new

particle formation (see e.g., Cai et al., 2021) and trimer formation can also tell us about the clustering and total levels of SA.



When the NPF event starts, the overall ion signal decreases, likely because ions are consumed by the new particles. During the most intense particle formation period when we can see particles around 10 nm and below, the chemical composition of ions is largely dominated by sulfuric acid. The fraction of sulfuric acid trimers compared to the dimer and monomer also seems to be higher than before the start of the event. This indicates that sulfuric acid is likely to play a role in land-influenced particle formation.

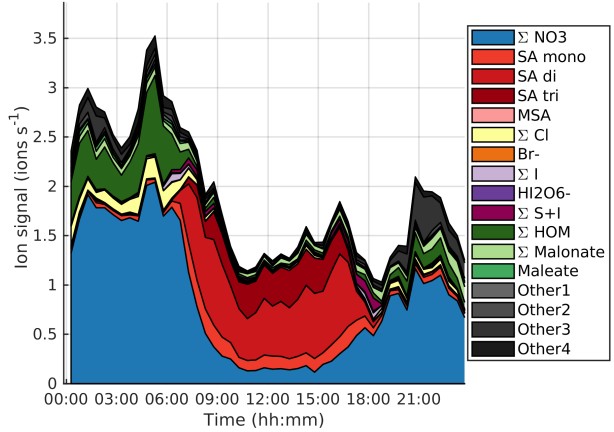

**Figure 9.** Time evolution of different ion groups during October 10th 2020.

The mode continues to grow until the end of the day although there are some fluctuations in the mean mode diameter. Since sulfuric acid signal goes down already around 18 h, sulfuric acid cannot be the only species growing the particles. In the evening, Other3 and different organic groups show an increase after the decrease of sulfuric acid signal. The fact that HOMs are observable both during the morning, which is after a different NPF event and in the evening of this NPF day, shows that HOMs were formed during the particle formation days even though they were not observed during the day, likely due to instrumental limitations. This indicates that they likely played a role in particle formation and growth over New Zealand. To fully confirm these interpretations, future measurements should also cover the chemical composition of neutral clusters and particles in larger sizes.

To get a better overview of how the chemistry affects particle formation, Figure 10 shows the average diurnal patterns of ion signals for NPF event and non-event days. Event days contain clear NPF event days (Class I and Class II days) as defined by the Dal Maso et al. (2005) criteria. The total ion signal reaches higher values on non-event days, which is expected as our previous work showed that the condensation sink is slightly higher on event days (Peltola et al., 2021) and this increases the losses of ions. Other differences between event and non-event days include more variable chloride signals and higher signals of organic species on event days, higher iodine oxide signals on non-event days and different relations of different sulfate species. On event days, chloride signals are relatively high in the morning and low in the evening while on non-event days the signal is relatively similar during the mornings and evenings. Events are likely favoured by clean chloride containing air coming from



the sea in the morning and the air crossing over land during the day. During the events the condensation sink increases and air masses age, leading to loss of the chloride signal.

The signal of HOMs is higher on event days compared to non-event days. Having higher HOM signals on event days
supports the assumption that HOMs can play a role in land-influenced NPF at Baring Head. Even though the total signal of ions of sulfuric acid is higher on non event days, likely due to the higher condensation sink observed on NPF days, the fraction of SA dimers relative to monomers is higher on event days (for whole day median $\frac{SA_{dimer}}{SA_{monomer}} = 0.54$ (25th-75th percentiles 0.16-2.58), during 10-15 h NZST 3.36 (2.50–4.82)) than on non-event days ($\frac{SA_{dimer}}{SA_{monomer}} = 0.43$ (0.15-1.95), during the day 2.74, (1.47-4.73)). Sulfuric acid dimer formation has been shown to be an indicator of NPF (Cai et al., 2021) and having more
dimers and trimers relative to monomers also indicates higher total sulfuric acid levels (Beck et al., 2022). This confirms the results of our case study that indicated that sulfuric acid is likely to play a role in NPF at Baring Head.

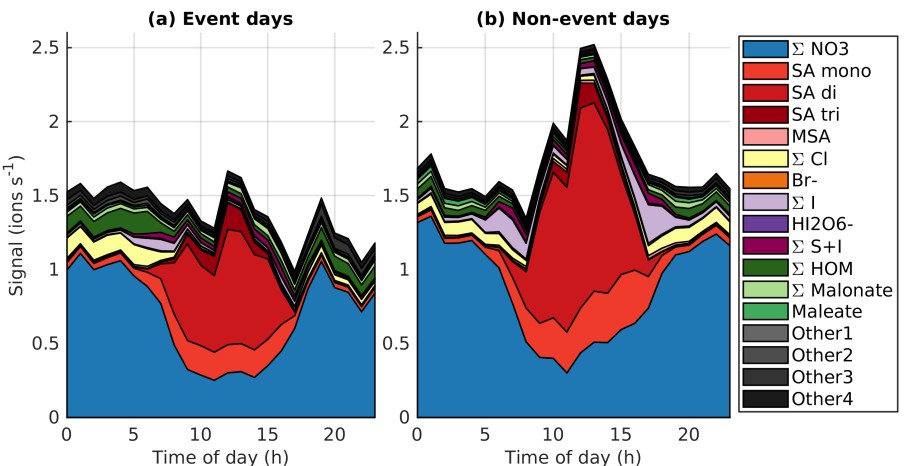

**Figure 10.** Average diurnal cycles of different chemical species on a) event days and b) non-event days.

To further understand the connections between new particle formation and different chemical species, we compared the Spearman's correlation coefficients between different ion signals, particle number concentrations and particle growth rates. Here, we focus on land-influenced data. Because the concentrations of both different chemical species and aerosol concentra-
tions have clear diurnal cycles and this could affect the interpretation of the results, we look at the correlations separately for morning (5-10 h, Fig. B1), daytime (10-15 h, Fig. B2), and nighttime (0-5 h, Fig. B3) data.

In the morning, around sunrise and the start time of earliest events (see, Peltola et al., 2021), 2-4 nm ions which can be seen as a first step of particle formation (Dada et al., 2018) correlate positively with all chemical species containing sulfur (sulfuric acid MSA and the S+I group, Fig. B1). SA di and SA tri also correlate with 2-4 nm ions during the day (Fig. B2). Weak positive
correlations can be seen also between sulfur species and N1-10 in the morning. These results support our earlier findings that sulfuric acid is likely one of the drivers of NPF over land. During the day, there is also a weak positive correlation between N1-10 and HOMs, despite the fact that HOM signals were low, because we likely observe only a small fraction of them during





the day. These results confirm the role of both sulfuric acid and HOMs in particle formation and show that different species can be important during different times of the day.

In larger size classes, N10-100 has only a weak positive correlation with Other3 group in the morning and SA trimer, malonate and Other3 groups during the day. N100 has positive correlation with malonate both during the morning and day. This is again confirming that organic species and sulfuric acid are important for particle formation over land. Most of the other correlations between particle number concentrations and different ion species are negative, which can be explained by higher losses of ions to the particle surface when particle concentrations are high.

With particle growth rates, the only significant positive correlations are found between Aitken mode growth rates and SA trimer during the morning (Fig. B1) and nucleation mode GRs and Other4 group and accumulation mode GRs and bisulfate ions and chloride during the day (Fig. B2). The Other4 group is likely to contain HOMs, indicating a role of organic species in the growth of nucleation mode particles.

At night (Fig. B3) no clear NPF was observed in land-influenced air and the correlations between the chemical species 450 and 2-4 nm ions and 1-10 nm particles are weak. Looking into the correlations can still shed a light on for example the growth processes. Concentrations of larger particles have positive correlations with malonate and Other3 groups, indicating that organic species or some unidentified compounds could play a role in particle growth to larger sizes overnight. Growth rates in all modes have negative correlations with chloride. This is opposite to what was seen during the day for accumulation mode growth rates. Chloride ion signal was typically higher during the night and higher in marine air masses, so negative 455 correlations with chloride likely show that nighttime growth was faster when the air had spent more time over land and the signals of marine species had decreased. Similar effects can be observed for bromide and MSA. To summarise, sulfuric acid and organic compounds are likely responsible for the new particle formation events that we observed in air masses crossing over land.

### 3.5.2 Marine new particle formation

Our previous work showed that in clean marine air, new particle formation should not be studied with the traditional criteria used for continental sites. Instead we focus on particle growth episodes and appearance of sub-10 nm particles. As an example day, we chose 15 October 2020. Almost all of this day was classified as clean marine air and we can see several growth events in the particle size distribution (Fig. 11a). The particle growth, as observed by the automatic method, first started after 6 h NZST and it can be observable in two different modes, one around 20-30 nm and another around 100 nm. We also show PSM data 465 for sub-10 nm particles instead of NAIS data, since NAIS seemed to underestimate particle concentrations which is especially problematic in the clean marine air where the concentrations are low. We can see that sub-10 nm particle concentrations are also elevated in the morning between 9-12 h and increase again towards the evening with a peak around 21 h, although the highest peak in the evening could be related to pollution since this time period contains some land-influenced periods as shown with white stripes in the particle size distribution. There are also several periods when the concentrations of 10 nm particles 470 in the SMPS data are elevated, both during the night before 5 h and during the morning around 9 h NZST. The air-mass back





trajectories for this day all come from the south and considering that for most of the time their altitudes are well below 1000 m, the air has likely been within the marine boundary layer (Fig. 11b).

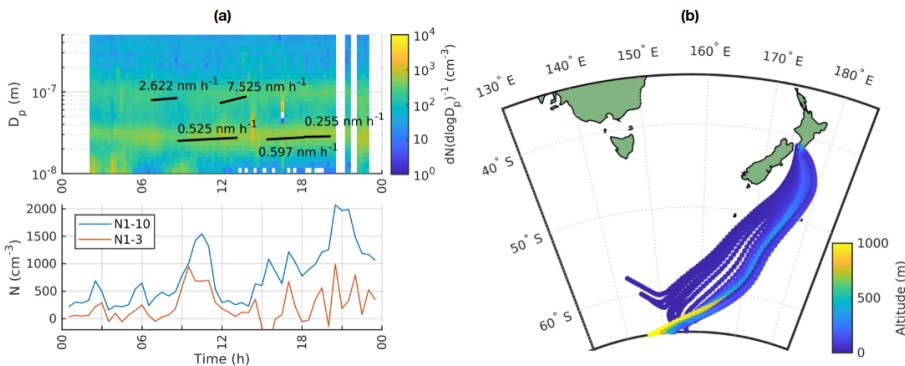

**Figure 11.** a) Upper panel shows the time evolution of particle size distribution in clean marine air with white vertical stripes with data missing corresponding to land-influenced periods and the lower panel shows the time evolution of N1-3 and N1-10 on 15 October 2020. b) Air-mass back trajectories coloured by altitude for the same day.

Figure 12 shows the time evolution of the chemical species during the same day. Around the start of the first growth episode after 6 h, we can observe nitrates, sulfuric acid, MSA, chloride and iodine oxides. Apart from lower iodine oxide signals

between 9-15 h, the observable species remain the same throughout the day even though their signals vary. Through 6-18 h, SA is the most abundant ion group and even though MSA is observable already during the day, its signal gets even higher in the evening when the SA signal decreases. This indicates that both SA and MSA were likely formed during the day by photochemistry, but SA might have masked some of the MSA signal during the day. This has been seen before by Tanner and Eisele (1991). SA and MSA are both condensable species and they can explain the observed particle growth at least partly.

This has been seen before in the Arctic by Beck et al. (2021). Iodine oxides were also observable during part of the growth and their potential to grow particles cannot be ruled out.

If we focus on the times when the sub-10 nm particle concentrations were higher (between 9-12 h and after 15 h), it is hard to say for certain which chemical species formed these particles. In the morning, sulfuric acid is the only clearly observable species that is known to participate in nucleation, but in the evening SA signals decrease and we can observe iodine oxides

which are also known to participate in particle formation.

To further relate the observed chemical species to aerosol formation, Figure 13 shows the diurnal variations of grouped ions in marine air when the number concentration of 1-10 nm particles is either below or above $500 \, \mathrm{cm}^{-3}$ at a given moment. Here, we follow the same logic as Peltola et al. (2021), and focus on 1-10 nm particles instead of NPF events since clear traditional NPF events were not observed in clean marine air. With the $500 \, \mathrm{cm}^{-3}$ limit, we have no chemical composition data around

3-5 h or 12-14 h where N1-10 would be above this limit. Decreasing the $500 \, \mathrm{cm}^{-3}$ particle limit to $100 \, \mathrm{cm}^{-3}$ or increasing it to $1000 \, \mathrm{cm}^{-3}$ influenced the availability of data at given times, but did not change the overall conclusions.





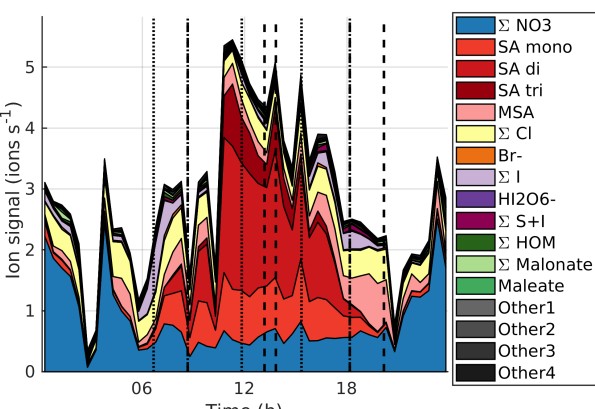

**Figure 12.** Variation of signals of different ions during one marine growth event on October 15th 2020. The dotted vertical lines show the start times of the growth events and the dashed lines show their end times.

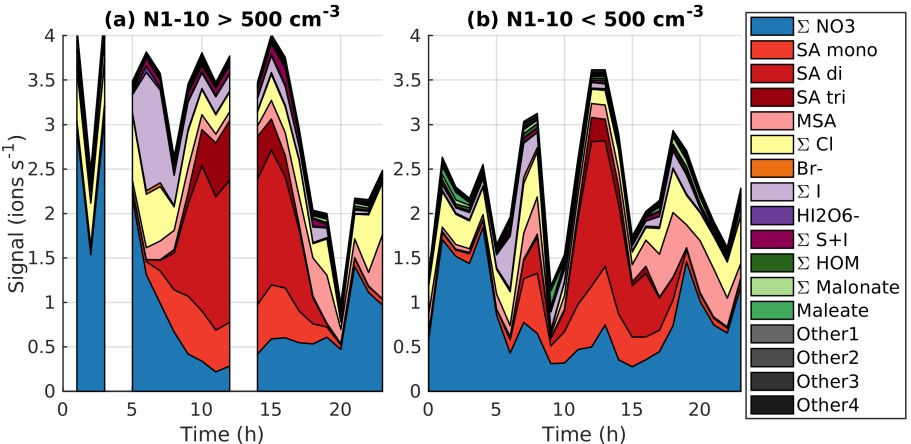

**Figure 13.** Average diurnal cycles in marine air when the number concentration of 1-10 nm particles is a) over or b) below $500 \ \mathrm{cm}^{-3}$.

The most notable differences between high and low N1-10 data seems to be higher signals of iodine oxide ions in the morning and higher SA trimer signals during the day when N1-10 is high. During high N1-10 days we also see slightly higher signals of compounds with both iodine and sulfur. Recent work has shown that iodine and sulfur species could form particles together (Ning et al., 2021) and our results indicate that this could be the case at Baring Head. Malonate seems more common during low N1-10 days although even then its signal is low. While the data are still not straightforward to understand, these results indicate that iodine oxides, sulfuric acid or species with both iodine and sulfuric acid could play a role in marine nucleation.

Finally, we looked into the correlations between different chemical species and aerosol data in marine air, as was done for the land-influenced data. Again, the data was divided to morning (5-10 h, Fig. B4), daytime (10-15 h, Fig. B5) and nighttime (0-5 h, Fig. B6) data. Compared to the land-influenced data, with the clean marine data we have fewer significant correlations since





there are less data. Now, in the morning 2-4 nm ions have a positive correlation only with the Other2 group (Fig. B4) and in the daytime data, the 2-4 nm ions have no positive correlations (Fig. B5). This is not surprising since the concentrations of these ions were very low in marine air. N1-10 has positive correlations with all the sulfuric acid signals and the S+I groups during both morning and day and in the morning also with HI2O6-. This supports our earlier results which indicated that sulfuric acid
and iodine oxides are likely important for particle formation in clean marine air.

For N10-100 we can observe positive correlation with MSA and Other4 group in the morning and sulfuric acid peaks during the day. This shows that sulfur species can also grow particles past 10 nm. N100 only has negative correlations with the groups containing iodine during the day and no significant correlation during the morning. Some possible explanations are that N100 acts as a sink for iodine oxides or that processes forming N100 are related to inhibiting formation of iodine oxides.

Nucleation and Aitken mode growth rates have positive correlations with malonate during the day, showing that organic species could potentially grow sub-100 nm particles in marine air. In nucleation mode, we also observe negative correlations with iodine oxides, SA trimer and species containing both iodine and sulfur. One possible explanation for this is that these species are consumed when particles grow. For daytime Aitken mode GRs, we also have positive correlations with bisulfate, chloride, and the Other4 groups whereas in accumulation mode all of these ions have negative correlations with the growth
rates. This shows that different mechanisms are important for the growth of particles in different sizes. In the morning, most of the correlations between growth rates and chemical species are insignificant due to the limited size of the data set.

During the night (Fig. B6), the 2-4 nm ions have no significant correlations with the different chemical species, but N1-10 has positive correlations with nitrates, bromide, iodine oxides, species with iodine and sulfur, maleate and Other4. This shows that the various different species can be involved in the nighttime formation of particles in marine air. For N10-100 there are
no significant correlations with the chemical groups and for N100, we have only negative correlations with some chemical species, which is logical since N100 can act as a sink for the ions. The growth rates also have no significant correlations since nighttime growth periods in marine air were not numerous.

Even though the chemical composition of ions can be complicated to interpret and measurements of total concentrations of these species would be useful, our data are valuable and provide insights on the chemical precursors responsible for particle
formation at Baring Head.

## 4   Conclusions

This article reported the measurements of chemical composition of ambient ions with an APi-TOF mass spectrometer at Baring Head station, in coastal New Zealand. This was the first time such measurements have been performed in the region. We discussed the most commonly observed chemical species and explored their behaviour and sources. Additionally we used
these data to study which chemical species act as precursors for new particle formation. The most abundant ions were nitrates during the night and sulfuric acid during the day in both marine and land-influenced air. Methane sulfonic acid, chloride, bromide and iodine oxides had clearly higher signals in marine air, which is reasonable since all of these species can come from marine sources. While methane sulfonic acid and chloride had daytime minimums and higher signals at night in marine



air, bromide and iodine oxides had morning and evening maxima. Highly oxygenated organic molecules had a clear nighttime maximum in land-influenced air and lower signal in marine air, showing that the precursors of these organic species likely originated from over land. Inspecting the seasonal cycles and geographical source areas showed that some compounds, such as methane sulfonic acid, were linked to marine biological activities.

While these data are complicated to interpret and with this data set we cannot quantitatively study the exact particle nucleation and growth mechanisms, we were able to identify species likely responsible for particle formation at Baring Head. For this we used case studies, correlation analysis, and comparing the levels of different chemical species during particle formation events and times when no particle formation was observed. In clean marine air masses, iodine oxides and sulfuric acid seemed to play an important role in particle formation, while sulfuric acid and potentially organic acids were important for particle growth. Observing iodine oxides at the station frequently indicates that they can be important for particle formation also in open ocean air masses and not only in coastal environments which have been the focus of iodine oxide particle formation studies in the past. In land-influenced air, sulfuric acid and organic compounds such as HOMs were important for particle formation which is in line with results observed at some continental sites in the past. Our results are highly valuable since the chemical composition of ions had not been measured in the region before and observations from coastal sites are overall rare.

In the future it would be good to expand the measurements to include measurements of the total concentrations of gases and chemical clusters using for example bromide chemical ionisation which is suitable for detection of not only sulfuric acid and key organic compounds but also iodine oxides, which can all be important for new particle formation at Baring Head. Simultaneous measurements of particle chemical composition in different sizes could further clarify the chemical processes controlling aerosol formation at Baring Head.

## 5   Data availability

The meteorological data can be downloaded from https://cliflo.niwa.co.nz/ and tide height data from https://www.linz.govt.nz/sea/tides/tide-predictions. The aerosol data is available on the AERIS data base (https://sea2cloud.data-terra.org/en/catalogue/), the CPC data is found at https://doi.org/10.25326/354, SMPS data at https://doi.org/10.25326/356 and PSM data at https://doi.org/10.25326/355. The APi-TOF data will be made available on AERIS data base before final publication.

*Author contributions.*   MP performed the APi-TOF measurements, analysed the data, and wrote the paper with contributions from all authors. MP and JT performed the aerosol measurements. SG processed the radon data. KS, CR and MH supervised the work.

*Competing interests.*   The authors declare that they have no conflict of interest.



*Acknowledgements.* These results are part of a project that has received funding from the European Research Council (ERC) under the European Union's Horizon 2020 research and innovation programme (Grant agreement No. 771369). The Sea2Cloud project is endorsed by SOLAS. The authors gratefully acknowledge the NOAA Air Resources Laboratory (ARL) for the provision of the HYSPLIT transport and dispersion model and READY website (https://www.ready.noaa.gov) used in this publication. Supporting measurements from the Baring

565   Head site are funded through NZ MBIE Strategic Science Investment Fund programme "Understanding Atmospheric Composition and Change".



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



## Appendix A: Peak list

**Table A1.** The list of the peaks and their masses used for the APi-TOF data in this study.

| Name | Mass (Th) | Name | Mass (Th) | Name | Mass (Th) | Name | Mass (Th) |
|------|-----------|------|-----------|------|-----------|------|-----------|
| Cl- | 34.969401 | unknown0071 | 130.971848 | unknown0044 | 270.917999 | unknown0083 | 339.024475 |
| NO2- | 45.993452 | unknown0055 | 130.993484 | unknown0045 | 271.001862 | unknown0084 | 339.072327 |
| ClH2O- | 52.979966 | unknown0072 | 142.967468 | unknown0046 | 271.009796 | C10H14O9NO3- | 340.052148 |
| CO3- | 59.985292 | unknown0073 | 143.000366 | (SO4)H2IO3- | 272.857144 | unknown0082 | 340.983337 |
| NO3- | 61.988366 | unknown0074 | 143.0065 | unknown0012 | 281.008209 | unknown0030 | 342.035004 |
| ClH4O2- | 70.990531 | unknown0004 | 154.010803 | unknown0011 | 281.00946 | unknown0081 | 345.019531 |
| Br- | 78.918886 | (NO3)C3H4O4- | 165.999325 | unknown0010 | 282.0177 | HI2O6- | 350.786805 |
| (NO3)H2O- | 79.998931 | IO3- | 174.889764 | unknown0013 | 282.983582 | unknown0064 | 355.049133 |
| CH3O3S- | 94.980839 | unknown0075 | 190.914418 | unknown0014 | 283.020813 | unknown0063 | 356.038696 |
| (SO4)H- | 96.960103 | H2IO4- | 192.900329 | unknown0015 | 284.013702 | unknown0062 | 357.039154 |
| (NO3)H4O2- | 98.009496 | unknown0041 | 192.942331 | (SO4)3H5- | 292.894863 | unknown0031 | 358.032379 |
| ClO2S- | 98.931302 | (SO4)2H3- | 194.927483 | unknown0078 | 294.890594 | unknown0065 | 359.029572 |
| CCl2H3N- | 98.964803 | unknown0076 | 196.925415 | unknown0017 | 297.018707 | unknown0066 | 359.069183 |
| C3H3O4- | 103.003682 | unknown0079 | 210.893204 | unknown0018 | 298.02536 | unknown0033 | 371.043579 |
| unknown0052 | 106.999748 | unknown0056 | 212.962232 | unknown0019 | 299.006287 | C10H14O11NO3- | 372.041978 |
| (SO4)O- | 111.947193 | unknown0008 | 217.005585 | unknown0020 | 299.018127 | unknown0034 | 373.01178 |
| (NO3)ClH2N- | 112.975943 | unknown0009 | 231.023146 | unknown0021 | 308.041504 | unknown0035 | 387.032532 |
| unknown0068 | 112.992399 | (SO4)CH3I- | 237.880225 | unknown0022 | 309.037079 | unknown0037 | 389.868835 |
| (SO4)H3O- | 114.970668 | unknown0057 | 254.964493 | unknown0023 | 313.013916 | unknown0038 | 389.929749 |
| C4H3O4- | 115.003682 | unknown0058 | 255.01454 | unknown0080 | 314.041618 | unknown0039 | 390.034332 |
| C4H6NOS- | 116.017559 | Cl4I- | 266.780431 | unknown0059 | 315.019989 | unknown0067 | 405.025024 |
| unknown0053 | 118.958336 | unknown0049 | 266.979061 | unknown0026 | 326.030151 | unknown0036 | 407.00293 |
| unknown0054 | 118.994736 | unknown0051 | 267.042408 | unknown0027 | 326.045349 | unknown0085 | 407.876007 |
| (NO3)2H- | 124.984009 | unknown0077 | 268.753937 | unknown0028 | 328.026489 | unknown0086 | 407.916199 |
| unknown0069 | 128.955978 | IOS4- | 270.78822 | unknown0060 | 329.018372 | unknown0087 | 407.967194 |
| unknown0070 | 128.988739 | unknown0043 | 270.873688 | unknown0061 | 331.02948 | | |



## Appendix B:  Additional figures

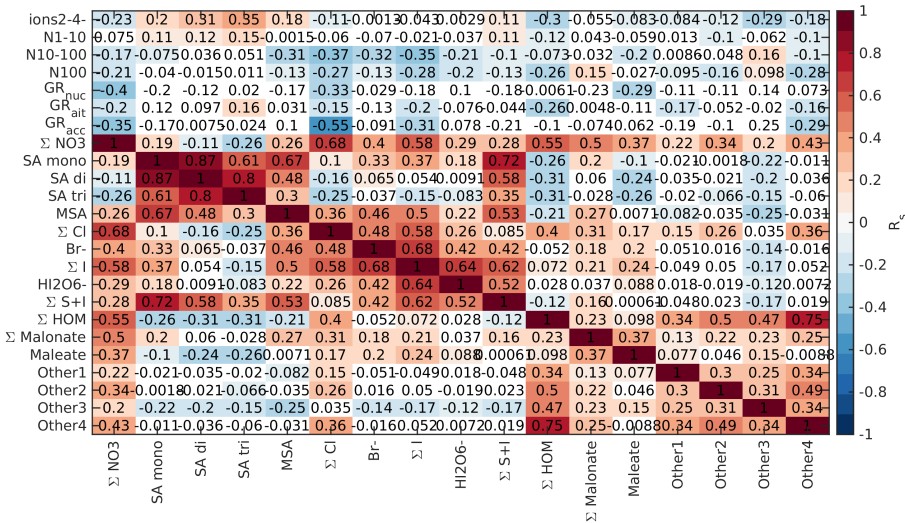

**Figure B1.** Spearman correlation coefficients between aerosol data and different chemical species in land-influenced air during the morning (5-10 h). Each square corresponds to the correlation coefficient between the variables on the x- and y-axes and the colour indicates the strength of the correlations with the square being white if p-value was above 0.05. The variables include negative 2-4 nm ions, particle number concentrations (N) in three different size bins (1-10, 10-100 and > 100 nm), particle growth rates (GR) in nucleation (< 25 nm), Aitken (25-100 nm) and accumulation (> 100 nm) modes and signals of different ions from the APi-TOF data

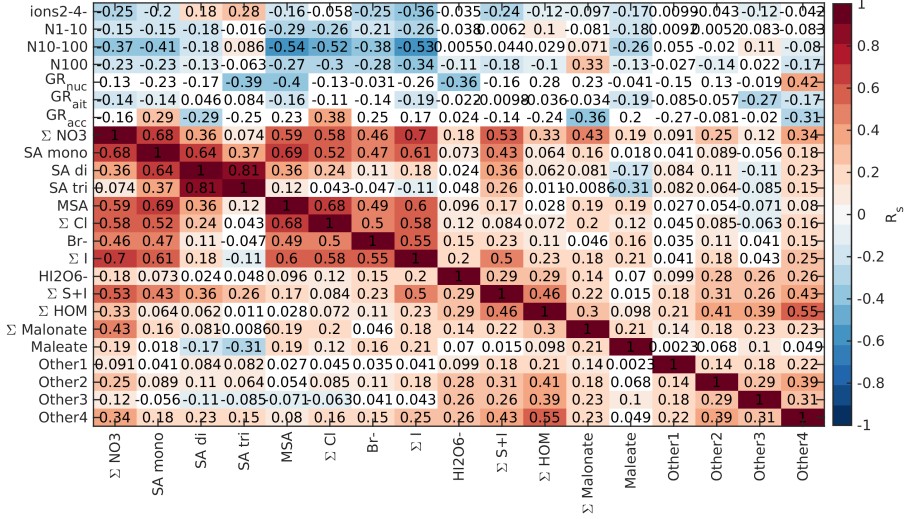

**Figure B2.** Spearman correlation coefficients between aerosol data and different chemical species in land-influenced air during the day (10-15 h). See Figure B4 for a full explanation.





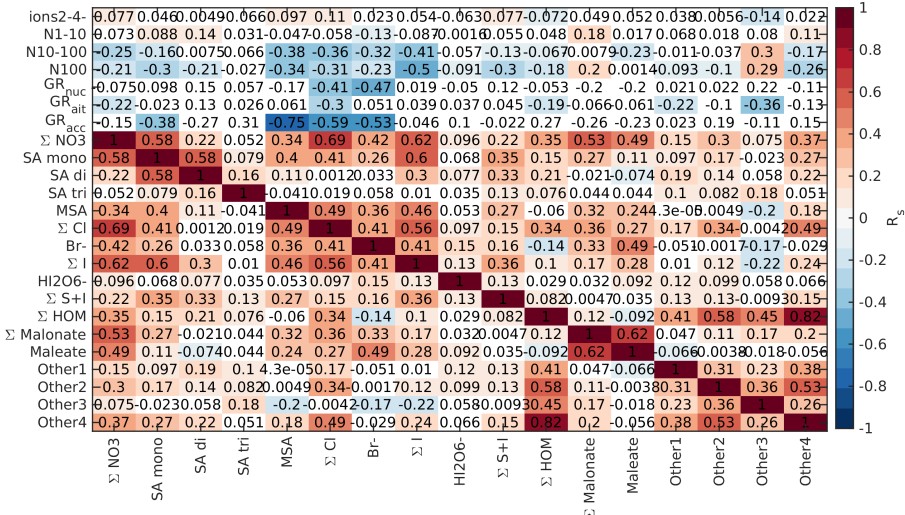

**Figure B3.** Spearman correlation coefficients between aerosol data and different chemical species in land-influenced air during the night (0-5 h). See Figure B4 for a full explanation.

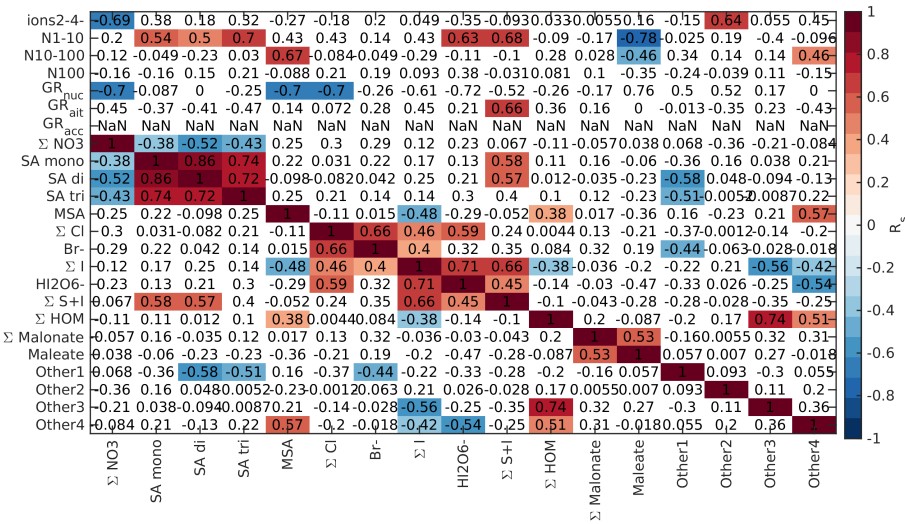

**Figure B4.** Spearman correlation coefficients between aerosol data and different chemical species in marine air during the morning (5-10 h). See Figure B4 for a full explanation.





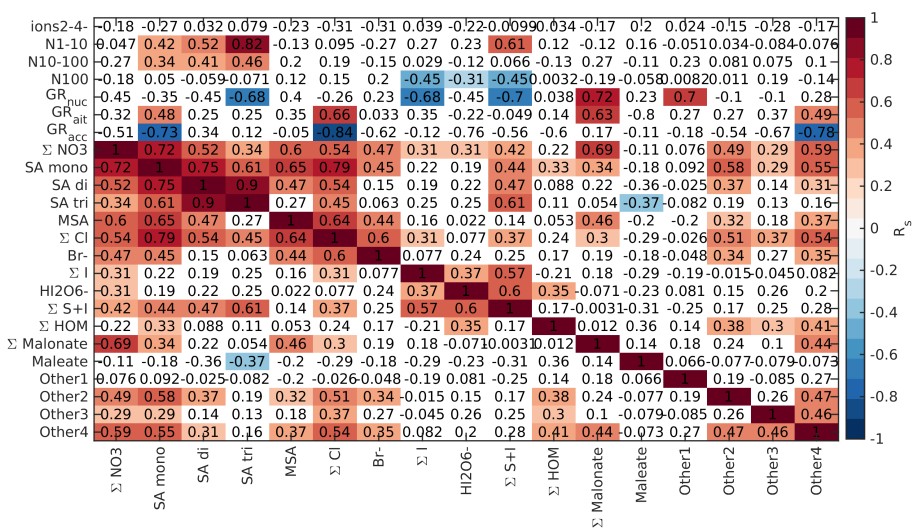

**Figure B5.** Spearman correlation coefficients between aerosol data and different chemical species in marine air during the day (10-15 h). See Figure B4 for a full explanation.

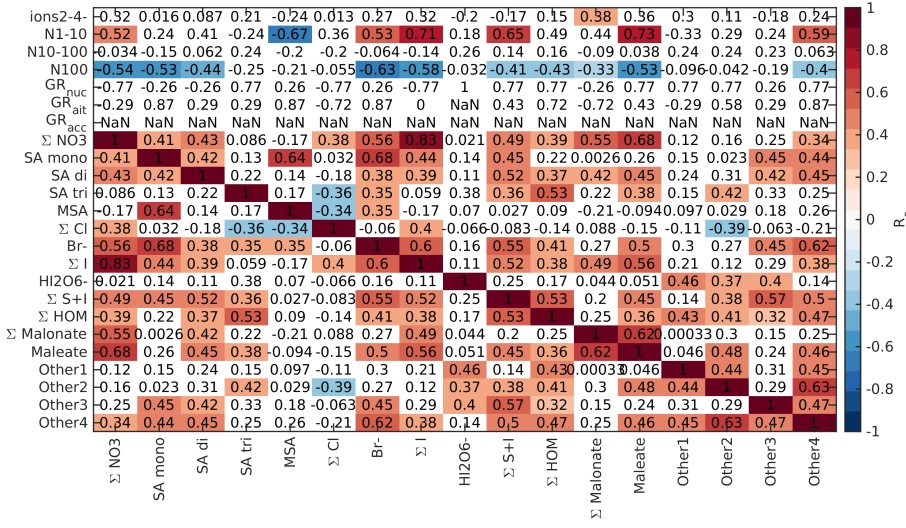

**Figure B6.** Spearman correlation coefficients between aerosol data and different chemical species in marine air during the night (0-5 h). See Figure B4 for a full explanation.