# Peer review of "Chemical precursors of new particle formation in coastal New Zealand"

_Atmospheric Chemistry and Physics, 2022_

## Referee Comment (RC1)

Peltola et al., 2022 represents the first measurements the composition of naturally charged ions from Baring Head, New Zealand. The paper addresses a very relevant scientific question of how aerosols are formed in a marine environment and adds context to the previous paper by the author (Peltola et al., ACP, 2022).

The manuscript introduces ion composition data from new areas of the world, which are in a key role when trying to reduce bias in aerosol formations studies, that mostly concentrate on studies in the boreal forest zone and Europe. Other than Antarctic and Bolivian measurements, these are one the first ambient observations from the Southern hemisphere. The concept is often used in aerosol formation studies but it is still rare to see actual field measurements and especially for such long time series of aerosol precursors (7 months of data). The authors characterize diurnal and seasonal cycles of air ion composition, different types of correlations and pointed out the source regions for these components. Where they could do much better is to conclude which components form molecular (ion) clusters and make more comparable correlations to nanoparticles by changing the size groups. This is something their data set is (or should) be capable of showing.

All together the paper has good scientific significance and quality and it is mostly presented in a well-structured way. In order to keep this as a scientific article I suggest more detailed analysis the results part. If you wish to keep the conclusions and results as they are now, I would suggest very minor corrections and a changing this article to a measurement report as at the current state it does not bring novel information on marine NPF to my opinion, but a lot of observations that should be published in ACP for sure. Suggestions to how to revise this scientific article are below in the more specific comments.

More specific comments:

L4: Ambient anions. You may use anions throughout the text also.

L35: Did you check your data if you can identify these compounds in your data set? I know Veres et a., 2020 used an iodine CIMS, but one can always look if they are ionized naturally.

L50-55: I suggest you add Beck et al., 2022 as citation in this section also since it introduces similar results from the area from two different sites.

L65: References within Bianchi et al., 2019 since it is a review paper, I would recommend to cite the original work if possible.

L75-76: Sulfate formation? Secondary sulfates (SA?), maybe I misunderstand this.

L80-82: Do you want to concentrate on marine NPF or both marine and land-influenced air? The introduction and your title concentrates very much on marine NPF but I think it would be valuable to include both since you have all the data you need to solve both. You could address the difference between land and marine air in the title even.

L93 / L100 and so on: Peltola et al., ACP, 2022 now, I assume. Great paper, I have to say, congratulations.

L102: What was the resolution of the device? (LTOF or HTOF?)

L104: Did you use 30 min averaging to investigate NPF events also? In clean locations, actually in most places, I use a much longer averages to increase the signal level. E.g. in Jokinen et al., 2018 paper from Antarctica, the mass defect plots are 180 min averages over the NPF event duration. I suggest you try the same to catch the possible higher SA oligomers and ammonia clusters.

L107: Increased? Did you have a different flow before?

L116: How many events (or any) did you catch during this 4.3 % of clean marine air when you had APi-TOF data?

L128: You used all data to gather the peak list in Table A1? It seems like very few peaks were present if this is the case.

L137: Is it an event day, non-event day? Since the charge is distributed to the strongest candidate, I would rather consider plotting an average night time and day time spectra since they have very specific features due to this.

L138: Can you differentiate $SO_3^-$ from $H_2ONO_3^-$? Water has a tendency in evaporating, but can be identified most of the time with nitrate ion.

L141: In Table A1 you list a peak "unknown0085" at 407.876007 Th and the exact mass of $NH_3(H_2SO_4)_3HSO_4^-$ is 407.888 Th. Is the mass accuracy sufficient to say that this peak is not ammonia-sulfuric acid-bisulfate cluster?

L145: There are multiple peaks in Table A1 that are marked as unidentified, however, I think a good amount of these peaks you can identify using the data in e.g. Ehn et al., 2012, Yan et al., 2016 PMF paper, Jokinen et al., PNAS, 2015 paper and supplementary figures that have many compositions of these peak, Bianchi et al., 2017 (https://doi.org/10.5194/acp-17-13819-2017) and many more if you wish. Loads of possibilities with this dataset!

L148-150: Clean air (air quality) can hold a lot of compounds in it like it is shown in Hyytiälä. Most importantly you seem to have weak sources of condensing vapours or sufficient sink for them before getting detected. Do you have an estimation of the losses in the inlet line or the transmission of the instrument? As you mention before, the APi-TOF has much lower detection threshold for ions than the the one coupled with chemical ionization, so Baccarini et al., 2021 probably had the same "problem" as you, not enough production of the compounds and higher detection limit.

L160 / Schemaballs: All data? Not NPF vs. non-NPF or night vs. day? It would be useful to compare night time and daytime data separately that are unfortunately not shown at the moment. To be honest, I have difficulties in reading the schemaballs with this much data and compounds in them. Statistically all the correlations seen are significant so this seem like a good way to group the data, for future I suggest getting to know PMF, that might help grouping compounds in another way. The correlations from which the data was now grouped is not shown, please show that data also or instead of these two plots.

L164: Is $H_2IO_4$ an ion missing its charge? Perhaps $H_2OIO_3^-$? I would also be specific in terminology when measuring ions: bisulfate (not always sulfuric acid) and so on.

L170 and Table 2: About grouping the compounds: Please correct the charge of ions (some are now neutral), e.g. $NO_3^-$. I would also go through the table and fix the compositions like in group 1: $H_2ONO_3^-$, $HNO_3NO_3^-$ (hydrogen does not have a negative charge), group 2: $H_2SO_4HSO_4^-$, $(H_2SO_4)_2HSO_4^-$ and so on, MSA: $CH_3SO_3^-$. I don't understand why the $HIO_3IO_3^-$ is in the different group due lower signal? It should not matter if you add these group compounds up. What is the value of having the dimer separately from the monomer and hydrated monomer? You probably also notice some similarities in the "Others". Most masses are uneven, meaning that they may contain a certain number of nitrogen atoms. If you are interested in digging deeper to this, I suggest you start by looking at the Yan et al., 2016 PMF paper more closely on organic nitrates. "Other 4" contains a very interesting peak that was connected to NPF in Hyytiälä years back (Kulmala et al., 2013, Science).

L198: Monoterpenes and isoprene have more sources than forests, but I might say they probably do originate from vegetation. You have the means to identify more peaks since you have a mass spectrometer, Sulo et al., 2021 also helps with peak lists.

L205: and charge is always distributed according to the compound's proton affinity! This really has a large effect on what you can detect. There might be dozens of compounds in the air that you just can't detect because SA takes away all you charge.

Fig 4: I really like this illustration, clear and informative. Nicely tells a story about HOM originating from land and MSA from the ocean, you must have some nice sources around.

L213: $NO_3^-$ or nitrate radical?

L225: Can it also be increased production, since MSA and SA have same marine source and MSA is highly abundant in your data set?

L234: Do the marine species (like $Cl^-$ or $Br^-$) have correlation with higher windspeeds (sea spray)?

L245: You can give the lifetime estimation since you have condensation sink available (Peltola et al., 2022). Lifetime is proportional to the CS.

L256: I assume this is iodic acid dimer, $HIO_3IO_3^-$.

L257: Are the species clusters or sulfur and iodine containing compounds?

L293: Bisulfates increased during Austral summer months in land influenced air, so how do you connect that to increased DMS emissions? Marine air masses did not show an increase in bisulfates. When the production is higher, you should be able to see the higher clusters forming also. Did you observe those?

L313: There are long term measurements available from SMEAR I, that is not so far north as Baccarini et al., measurements from the Arctic if you want to compare to a clean continental area (Jokinen et al., 2022, https://doi.org/10.5194/acp-22-2237-2022).

L317: What is the temperature during winter? Lower temperature favours condensation (lower vapour pressure). CLOUD experiment has done temperature runs with HOMs if you want to have a look, e.g. Frege et al., 2018 (https://doi.org/10.5194/acp-18-65-2018). It would be interesting to see the meteorological data in this manuscript also.

L344: If DMS oxidation is the most important source of nss-$SO_4$, then why you link bisulfates with mostly $SO_2$ pollution and not DMS oxidation? Can you estimate how much of bisulfate dimer would come from DMS oxidation and how much from pollution? Are the source regions same for the monomer and dimer?

L352: What is the lifetime of DMS? Can it be transported to your site and get oxidized in the vicinity of your sampling site?

Fig 8A: Could you insert more ticks to see the diameter range better? And please add number concentration of particles like you have in Fig 11.

Fig 9, 10, 12,13: I would normalize the ion signals with TIC or just a use a diurnal plot of signals in order to see the increasing or decreasing signals. How many events in total (N) are depicted as NPF and non-events?

L406: The instrument measures naturally charged ions, so in that sense it is not the limitation of the instrument but actual phenomenon in the atmosphere. Bisulfates are produced during the day and (unfortunately) take the charge and hinder the identification of processes happening in the background.

L440-450: How about correlation with formation rates? That could be better be suitable to particle numbers.

Fig 11. I don't recommend using the size class 1-10 nm (if from NAIS), since the size classes 0.8-2nm (small ions) and 2-7 nm (intermediate ions) have very different dynamics and behaviour, see Manninen et al., 2016, fig 17. Therefore, I suggest you always separate them. The ones over 7 nm (large ions, 7-20 nm) should also be in different size class than the smaller ones. Make sure you depict the instruments used for the measurement in the figure caption.

(Manninen et al., 2016, https://amt.copernicus.org/articles/9/3577/2016/amt-9-3577-2016.pdf)

L478: Did you have solar radiation measurements? You could show the link between the formation of precursors and radiation (or just use the daylight times)

L482-485: I suggest you take the whole NPF period where small particles are formed, from 9-12 AM, and average it in one spectrum. Make a mass defect plot to interpret further, they are extremely useful tools for this type of analysis. I don't know what you want to say with the bisulfate signal decrease in the evening, since the formation event happens in the morning-noon?

L488: If you did the analysis based on the ion classes by Manninen et al., 2016, would this conclusion be different?

L495: I recommend you to look at mass defect plots and compare e.g. events to non-events, before and during event or marine to over land air masses.

L501-505: Again, the ion size groups are selected a bit differently. 2-4 and <10 nm. I suggest using the standard operation procedure recommendations by Manninen et al., 2016 (0.8-2, 2-7 and 7-20 nm) if reporting NAIS data. This way the data will be further comparable with ACTRIS data also.

L523: Do you have values of what is the fraction of ion induced nucleation at your site? The APi-TOF should be capable of revealing clustering of iodate or bisulfates, you might need to average much longer times than 30 minutes to build up signal from NPF times and I suggest looking into mass defect plot or plotting spectra during event and non events on top of each other in order to find the differences.

L548: concentration of neutral aerosol precursors? Total concentration of gases is a wide suggestion, but naturally recommended.

Thank you for introducing the measurements to me and best of luck finishing the manuscript!

---

## Referee Comment (RC2)

**Review comments: Chemical precursors of new particle formation in coastal New Zealand, M. Peltola et al., 2022**

This manuscript presents ion measurements from a remote site in the southern hemisphere using an atmospheric pressure interface time of flight spectrometer (APi-ToF) with no chemical ionisation inlet to capture ambient anions. This is an extensive dataset spanning seven months, and the accompaniment to a previous high quality paper published by the same authors (Peltola et al., 2022). The subject is of great scientific interest as it presents a comprehensive dataset in a highly unstudied area. The manuscript is well written, and figures are beautifully presented but would be greatly strengthened by some further analysis. Some key calculations can be done from the size distribution data (formation rates and ion-ion recombination rates at low diameters), some ion signals can possibly be assigned formulae with some careful thought, and if the averaging time is increased, some more mechanistic insight may be possible from the mass spectra. I highly recommend publication in ACP once a few comments are addressed.

**Specific comments**

Line 10: Should this say bisulphate, rather than sulphate? Or maybe sulphur-containing ions?

Line 69: If mentioning the sea surface microlayer, maybe reference (Mungall et al., 2017)

Line 98: It would be nice to have some more details here about the running conditions for the PSM, I presume it was used in scanning mode, how long did each scan take, and how was the data inversion performed? Chan et al., 2020 presents four different techniques for inverting this data that give slightly different results. Was there any data pre-processing?

Line 116: How many NPF events fall into this 4.3% of data coverage?

Line 117: How does this compare to the Potential Source Contribution Function? I.e., as discussed by (Fleming et al., 2012).

Line 130, Figure 2 & 3: These figures are very visually striking, but to me it is quite difficult to see the correlation between most of these species unless the correlations are particularly positive or negative. Would it make sense to reduce the alpha for lines corresponding to R values close to zero perhaps, as we are mostly concerned with stronger correlations?

Line 139: "Pure sulfuric acid clusters were detected up to the trimer". Do you mean sulphuric acid-bisulphate clusters? Also, is this referring to your dataset here, or Junninen et al.?

Line 153: It would be nice to say how you went about deciding what peaks to assign formulae to, and what could not be fit. What possible combination of atoms did you look at? What error (i.e. in ppm) did you deem acceptable?

Line 171: Why is this a good reason to put them in another group? Do you infer that $HI_2O_6^-$ has a different source than the other two iodine compounds from this?

Line 181: I'm not sure I understand the rationale here. Peaks in group "Other1" and "Other3" correlate with each other, but why is that important for the main thrust of the paper (aerosol formation?), especially when these contain 2 and 3 peaks, respectively. What counts as a "strong correlation" here?

I understand the grouping of "Other2" and "Other4" as they correlate with aerosol concentrations (I presume this means total number concentrations from your CPC? Or is this N100 from integrating across the size distribution measurements). I will also echo the other reviewer here and suggest digging deeper to try and assign more of these mass spectral peaks. For example, your "Other4" group contains peaks which very well could be of the formulae $C_xH_yO_zN_1NO_3^-$. Some quick calculations

show me that the peak at 339.024 m/Q would be about 150 *ppm* away from $C_{10}H_{15}O_8NNO_3^-$, and the peak at 373 about 150 *ppm* away from $C_{10}H_{17}O_{10}NNO_3^-$, the same with the two other ions. This discrepancy in mass could easily be mass calibration related perhaps

Line 194: It would be nice to substantiate the claim about halogen anions from the ocean with a reference

Line 213: Are you confusing the $NO_3$ radical with the nitrate anion here?

Line 225: I'm not sure you can substantiate this claim with your data here, how do you know this is explained by a higher condensation sink, rather than a lower source strength? The CS has been calculated for this dataset (Peltola et al., 2022). It might be helpful to use this here.

Line 235: Again, a reference r.e. these sources would be nice

Line 256: As this discussion doesn't add much to the overall discussion of iodine oxides, why not include it with the sum of iodine anions?

Line 266: I am not sure organosulphate is the correct term for a HOM-bisulphate cluster as this usually refers to molecules with a $R$-$SO_4^-$ functional group

Line 282: This nighttime peak in Other4 is somewhat consistent again with what was observed for Organonitrate-nitrate anion clusters in previous work (Bianchi et al., 2017)

Figure 6: Why are groups Other1-4 not included here?

Line 331: Just a note on this section: it may be useful also to try something like the potential source contribution function (PSCF) to simply identify the regions leading to the highest ion signals. This will exclude some of your data and perhaps highlight just the strongest source regions.

Line 340: The $H_2SO_4HSO_4^-$ cluster is a good indicator of NPF in Beijing, but what about this dataset?

Line 379: I think it should be possible to see some larger clusters during NPF events in the mass spectra, even with a suboptimal instrument tuning. Is it possible to average across the entire NPF event? Possibly when the ion concentrations from the NAIS are elevated. This may help the data interpretation somewhat

Figures 9, 10, 12, 13: It might be nice to see these in a non-stacked fashion to simply see how, for example, $H_2SO_4HSO_4^-$ behaves more clearly. These could simply go in the appendix.

Line 427: Why not investigate the correlation with the formation rates? The $J_{10}$ is available from the previous publication from this dataset, and the formation rate at lower sizes can be calculated from the available NAIS and PSM data, better yet, the ion-ion recombination rates can be calculated, which fit very nicely with the APi-ToF measurements. You may then find much better correlations with your ion signals.

**References**

Bianchi, F., Garmash, O., He, X., Yan, C., Iyer, S., Rosendahl, I., Xu, Z., Rissanen, M. P., Riva, M., Taipale, R., Sarnela, N., Petäjä, T., Worsnop, D. R., Kulmala, M., Ehn, M., and Junninen, H.: The role of highly oxygenated molecules (HOMs) in determining the composition of ambient ions in the boreal forest, Atmos. Chem. Phys., 17, 13819–13831, https://doi.org/10.5194/acp-17-13819-2017, 2017.

Chan, T., Cai, R., Ahonen, L. R., Liu, Y., Zhou, Y., Vanhanen, J., Dada, L., and Chao, Y.: Assessment of particle size magnifier inversion methods to obtain particle size distribution from atmospheric measurements, Atmos. Chem. Phys. Discuss., 1–21, 2020.

Fleming, Z. L., Monks, P. S., and Manning, A. J.: Review: Untangling the influence of air-mass

history in interpreting observed atmospheric composition, Atmos. Res., 104–105, 1–39, https://doi.org/10.1016/j.atmosres.2011.09.009, 2012.

Mungall, E. L., Abbatt, J. P. D., Wentzell, J. J. B., Lee, A. K. Y., Thomas, J. L., Blais, M., Gosselin, M., Miller, L. A., Papakyriakou, T., Willis, M. D., and Liggio, J.: Microlayer source of oxygenated volatile organic compounds in the summertime marine Arctic boundary layer, Proc. Natl. Acad. Sci. U. S. A., 114, 6203–6208, https://doi.org/10.1073/pnas.1620571114, 2017.

Peltola, M., Rose, C., Trueblood, J. V., Gray, S., Harvey, M., and Sellegri, K.: New particle formation in coastal New Zealand with a focus on open-ocean air masses, Atmos. Chem. Phys., 22, 6231–6254, https://doi.org/10.5194/acp-22-6231-2022, 2022.

---

## Author Response (AR1)

*We thank the reviewers for their helpful feedback. Please find our responses below in grey italics. Please note that in addition to taking into account the reviewers' comments, we changed 'iodine oxides' to 'iodine oxoacids' throughout the text, according to a comment from a colleague (Xu-Cheng He) since that is a more correct term. We also added the Maori name of Wellington to the text since that was required in the typesetting of our previous manuscript (Peltola et al., 2021).*

Peltola et al., 2022 represents the first measurements the composition of naturally charged ions from Baring Head, New Zealand. The paper addresses a very relevant scientific question of how aerosols are formed in a marine environment and adds context to the previous paper by the author (Peltola et al., ACP, 2022). The manuscript introduces ion composition data from new areas of the world, which are in a key role when trying to reduce bias in aerosol formations studies, that mostly concentrate on studies in the boreal forest zone and Europe. Other than Antarctic and Bolivian measurements, these are one the first ambient observations from the Southern hemisphere. The concept is often used in aerosol formation studies but it is still rare to see actual field measurements and especially for such long time series of aerosol precursors (7 months of data). The authors characterize diurnal and seasonal cycles of air ion composition, different types of correlations and pointed out the source regions for these components. Where they could do much better is to conclude which components form molecular (ion) clusters and make more comparable correlations to nanoparticles by changing the size groups. This is something their data set is (or should) be capable of showing.

All together the paper has good scientific significance and quality and it is mostly presented in a wellstructured way. In order to keep this as a scientific article I suggest more detailed analysis the results part. If you wish to keep the conclusions and results as they are now, I would suggest very minor corrections and a changing this article to a measurement report as at the current state it does not bring novel information on marine NPF to my opinion, but a lot of observations that should be published in ACP for sure. Suggestions to how to revise this scientific article are below in the more specific comments.

*While we understand why you think this data set should be capable of doing more, unfortunately we are not sure if it is capable of doing as much as one would wish. Even with long averaging times we cannot find any peaks with masses above ~400 Th and this makes it hard to draw any definitive conclusions on particle formation pathways. We think that even though we are not able to point out any specific aerosol formation pathways, this paper should remain as a research article since we are not only reporting new data, but also analysing and discussing the data extensively and showing which compounds are most likely to be involved in new particle formation in this region that had no previous similar measurements. Similar work in marine air is very scarce and often focused on coastal regions, meaning that even if our results are not surprising as such, they do bring new information about marine new particle formation.*

More specific comments:
L4: Ambient anions. You may use anions throughout the text also.
*We changed ions to anions throughout the text where applicable.*
L35: Did you check your data if you can identify these compounds in your data set? I know Veres et a., 2020 used an iodine CIMS, but one can always look if they are ionized naturally.

*We did not specifically check for this compound, but we did not find a peak around mass 108, so we assume that it was not observable in this data set.*

L50-55: I suggest you add Beck et al., 2022 as citation in this section also since it introduces similar results from the area from two different sites.

*We added a reference to Beck et al. (2021) next to Baccarini et al. (2020) on line 50*

L65: References within Bianchi et al., 2019 since it is a review paper, I would recommend to cite the original work if possible.

*We added a reference Ehn et al. (2014).*

L75-76: Sulfate formation? Secondary sulfates (SA?), maybe I misunderstand this.

*We changed 'sulfate to 'secondary sulfate aerosol' for clarity.*

L80-82: Do you want to concentrate on marine NPF or both marine and land-influenced air? The introduction and your title concentrates very much on marine NPF but I think it would be valuable to include both since you have all the data you need to solve both. You could address the difference between land and marine air in the title even.

*We show results for both marine and land-influenced air, but we wanted to highlight the marine part in the title since marine measurements are more scarce, which makes this research more novel.*

L93 / L100 and so on: Peltola et al., ACP, 2022 now, I assume. Great paper, I have to say, congratulations.

*Yes, it was not yet published at the time we submitted this paper. We have now updated the reference. Thank you!*

L102: What was the resolution of the device? (LTOF or HTOF?)

*It was an HTOF, we added this to the text.*

L104: Did you use 30 min averaging to investigate NPF events also? In clean locations, actually in most places, I use a much longer averages to increase the signal level. E.g. in Jokinen et al., 2018 paper from Antarctica, the mass defect plots are 180 min averages over the NPF event duration. I suggest you try the same to catch the possible higher SA oligomers and ammonia clusters.

*No, we used longer times, (varying around 6-24 h) to identify peaks. 30 min was just used to create the time series.*

L107: Increased? Did you have a different flow before?

*We mean increased compared to the < 1 lpm flow that the instrument takes in. We replaced 'to the instrument' by 'in this inlet' for clarity.*

L116: How many events (or any) did you catch during this 4.3 % of clean marine air when you had APi-TOF data?

*None of the clear regional events were in marine air.*

L128: You used all data to gather the peak list in Table A1? It seems like very few peaks were present if this is the case.

*Yes, we went through all of the data to create the peak list, although it is possible that we missed some peaks. It is true that the peak list is not very long, but even for most of the peaks in the list the signal was rather weak. The quality of this data set is most likely not as high as the reviewer is used to, potentially due to for example poor tuning or non-optimal (smaller diameter than what is typically used) inlet.*

L137: Is it an event day, non-event day? Since the charge is distributed to the strongest candidate, I would rather consider plotting an average night time and day time spectra since they have very specific features due to this.

*This data is from 24.8.2020, which was classified as an undefined day. We added a version with daytime and nighttime data separated to the appendix (see below) and added text 'The same data separated to day- and night time can be found in Figure B1.' to line 137.*

[Figure]

**Figure B1.** *Example mass spectrum from 24 August 2020 divided for a) daytime (9-15 h) and b) nighttime (0-5 h).*

L138: Can you differentiate SO3- from H2ONO3-? Water has a tendency in evaporating, but can be identified most of the time with nitrate ion.

*We checked this briefly for some day time data where sulfuric acid was abundant and no peak was observable at the mass of SO3-. The mass difference between SO3- and H2ONO3- is so large that we assume they should be separable.*

L141: In Table A1 you list a peak "unknown0085" at 407.876007 Th and the exact mass of NH3(H2SO4)3HSO4- is 407.888 Th. Is the mass accuracy sufficient to say that this peak is not ammoniasulfuric acid-bisulfate cluster?

*You are right that this is likely the composition of this peak, thank you for pointing this out! We looked at one day during which the signal of this peak is high and at this time the error for the peak was 15. ppm, which we would consider acceptable at this mass, since our mass calibration was made using lower masses. This peak also has positive correlations with the bisulfate peaks when looking at the whole time series which supports this peak identification. We replaced the text 'saw no sulfuric acid clustered with ammonia' on line 140 by 'only saw one peak with sulfuric acid clustered with ammonia (NH3(H2SO4)3HSO4-, in the peaklist unknown0085)' to take this into account.*

L145: There are multiple peaks in Table A1 that are marked as unidentified, however, I think a good amount of these peaks you can identify using the data in e.g. Ehn et al., 2012, Yan et al., 2016 PMF paper, Jokinen et al., PNAS, 2015 paper and supplementary figures that have many compositions of these peak, Bianchi et al., 2017 (https://doi.org/10.5194/acp-17-13819-2017) and many more if you wish. Loads of possibilities with this dataset!

*It is true that it would most likely be possible to identify some of these peaks. However, we did already go through the papers you mention when analyzing the data, but the identification was still challenging. While with some work we could potentially identify a few more peaks, we would prefer not going back to modifying the peak list since running it for the whole data set would take a long time and would not necessarily bring that much new*

*information to the paper since the number of peaks is quite small and we have already identified many compound groups.*

L148-150: Clean air (air quality) can hold a lot of compounds in it like it is shown in Hyytiälä. Most importantly you seem to have weak sources of condensing vapours or sufficient sink for them before getting detected. Do you have an estimation of the losses in the inlet line or the transmission of the instrument? As you mention before, the APi-TOF has much lower detection threshold for ions than the the one coupled with chemical ionization, so Baccarini et al., 2021 probably had the same "problem" as you, not enough production of the compounds and higher detection limit.

*Unfortunately we do not have an estimation for the losses or transmission.*

L160 / Schemaballs: All data? Not NPF vs. non-NPF or night vs. day? It would be useful to compare night time and daytime data separately that are unfortunately not shown at the moment. To be honest, I have difficulties in reading the schemaballs with this much data and compounds in them. Statistically all the correlations seen are significant so this seem like a good way to group the data, for future I suggest getting to know PMF, that might help grouping compounds in another way. The correlations from which the data was now grouped is not shown, please show that data also or instead of these two plots.

*Yes, these plots use all times of day. Originally we made these plots also for different times of the day and used those when grouping the data as well. Here they were left out since we already show the correlation matrices (Figures B1-B6) for different times of the day and they show the same information, just with less compounds and in a more readable format. Since there are over hundred variables (= over 10000 correlations) and the information that we considered the most important is already on the matrices, we would prefer not showing all the correlations. It is true that PMF would be a great option in the future.*

L164: Is H2IO4 an ion missing its charge? Perhaps H2OIO3-? I would also be specific in terminology when measuring ions: bisulfate (not always sulfuric acid) and so on.

*Yes, we added the charge. It is true that it is most likely H2OIO3-. We changed sulfuric acid to bisulfate anion on this line.*

L170 and Table 2: About grouping the compounds: Please correct the charge of ions (some are now neutral), e.g. NO3-. I would also go through the table and fix the compositions like in group 1: H2ONO3-, HNO3NO3- (hydrogen does not have a negative charge), group 2: H2SO4HSO4-, (H2SO4)2HSO4- and so on, MSA: CH3SO3-. I don't understand why the HIO3IO3- is in the different group due lower signal? It should not matter if you add these group compounds up. What is the value of having the dimer separately from the monomer and hydrated monomer? You probably also notice some similarities in the "Others". Most masses are uneven, meaning that they may contain a certain number of nitrogen atoms. If you are interested in digging deeper to this, I suggest you start by looking at the Yan et al., 2016 PMF paper more closely on organic nitrates. "Other 4" contains a very interesting peak that was connected to NPF in Hyytiälä years back (Kulmala et al., 2013, Science).

*The group names are just names that we tried to keep simple, so when there are more than one compound as indicated by the sum sign, we did not mark the charge. We fixed the charges in the table. The idea in having the dimer HIO3IO3- in a separate group was that it might indicate higher iodine oxoacid concentrations and potentially act as an indicator for new particle formation since this is something that has been seen for sulfuric acid. Yes, it is true that for example peaks in group Other4 likely have two nitrogens. We have added a mention of the possible chemical composition of the peaks in this group to the text (see reply to the other reviewer.)*

L198: Monoterpenes and isoprene have more sources than forests, but I might say they probably do originate from vegetation. You have the means to identify more peaks since you have a mass spectrometer, Sulo et al., 2021 also helps with peak lists.

*We changed 'forests' to 'vegetation'. We did use the paper by Sulo et al. for the peak identification and for example the peak of C10H14O9(NO3-) is also predominant in their work. However, from time to time this paper was confusing since not all the given masses match the compounds, for example the peak at 339 Th should probably be C10H15N2O11–, not C10H15O8N2(NO3-).*

L205: and charge is always distributed according to the compound's proton affinity! This really has a large effect on what you can detect. There might be dozens of compounds in the air that you just can't detect because SA takes away all you charge.

*Yes, this is true.*

Fig 4: I really like this illustration, clear and informative. Nicely tells a story about HOM originating from land and MSA from the ocean, you must have some nice sources around.

*Thank you.*

L213: NO3- or nitrate radical?

*We might have been a bit confused here, we thought the diurnal cycles of the radical and the anion would be similar. We replaced the text 'This is because during the day, NO3 is destroyed by photolysis  (e.g., Wayne et al., 1991).'  by 'One possible daytime loss term of nitrate ions is the loss of charge to sulfuric acid. This has been seen before at other measurement sites  (Eisele and Tanner, 1990; Yan et al., 2018).'*

L225: Can it also be increased production, since MSA and SA have same marine source and MSA is highly abundant in your data set?

*Yes, this is possible. We added text '  both higher sources of SA in marine air and' before 'smaller losses'.*

L234: Do the marine species (like Cl-  or Br-) have correlation with higher windspeeds (sea spray)?

*Yes, in marine air Cl- had correlation coefficient of 0.17 and Br- 0.36 (for both p< 0.05).*

L245: You can give the lifetime estimation since you have condensation sink available (Peltola et al.,2022). Lifetime is proportional to the CS.

*We added text 'In marine air the median condensation sink (see work by Peltola et al. 2022 for the data) for sulfuric acid is $4.7 \cdot 10^{-4}$, while in marine air it is only $2.8 \cdot 10^{-4}$. We can thus assume that the median lifetime of sulfuric acid in land-influenced air is 59% of its median lifetime in marine air.' after line 246.*

L256: I assume this is iodic acid dimer, HIO3IO3-.

*Yes, you are right.*

L257: Are the species clusters or sulfur and iodine containing compounds?

*We do not know exactly, since we are not experts in chemistry and we have not found these peaks reported before.*

L293: Bisulfates increased during Austral summer months in land influenced air, so how do you connect that to increased DMS emissions? Marine air masses did not show an increase in bisulfates. When the production is higher, you should be able to see the higher clusters forming also. Did you observe those?

*As we tried to explain in the text, one possibility is that DMS emissions and sulfuric acid formation increased also in marine air, but when they increased, the condensation sink also increased and in marine air the relative increase in condensation sink would have been larger and hindered the build up of sulfuric acid levels. The marine seasonal cycle is also more uncertain than the land-influenced cycle since we have clearly less marine data.*

*We also know based on the MSA data in this work (clear springtime increase in both air mass classes) and previous work at the station (particulate non sea salt sulfates, Li et al., 2018) that there is a springtime maximum in DMS emissions, MSA, and secondary sulfate aerosol formation, so it would be logical that the springtime increase we observe in bisulfate signals is related to this.*

*Both bisulfate anion alone and its clusters with one or two sulfuric acid molecules also have their highest signal in the spring-summer. So yes, we also observed more of the higher clusters during the spring-summer maximum. In marine air the highest median value of all different bisulfate clusters is observed in October and lowest in December, so some sort of a spring maximum is observed, but there is so little data that the trends between other months are less clear.*

L313: There are long term measurements available from SMEAR I, that is not so far north as Baccarini et al., measurements from the Arctic if you want to compare to a clean continental area (Jokinen et al., 2022, https://doi.org/10.5194/acp-22-2237-2022).

*Yes, that is an interesting dataset and they seem to have relatively high iodic acid concentrations in April, but since wintertime data is missing, it is hard to say if it is in agreement with the hypothesis mentioned here.*

L317: What is the temperature during winter? Lower temperature favours condensation (lower vapour pressure). CLOUD experiment has done temperature runs with HOMs if you want to have a look, e.g. Frege et al., 2018 (https://doi.org/10.5194/acp-18-65-2018). It would be interesting to see the meteorological data in this manuscript also.

*This is a good point, but the median temperature in June-August during our measurements was 10.6 °C which is only 2.05 °C lower than the median temperature of the whole measurement period, so we assume that the effect of temperature on HOM formation would not be too high. Even though we did look into the effect of meteorological conditions on the signals of different anions, we decided to keep the meteorological data out of this manuscript since even without that data, the manuscript is rather long and many of the connections with meteorological data are complicated to interpret.*

L344: If DMS oxidation is the most important source of nss-SO4, then why you link bisulfates with mostly SO2 pollution and not DMS oxidation? Can you estimate how much of bisulfate dimer would come from DMS oxidation and how much from pollution? Are the source regions same for the monomer and dimer?

*This is a good point, we replaced the text 'One possible reason for this is the transport of SO2 from Australia' by 'Possible reasons for this include higher sources of DMS in this direction and the transport of SO2 from Australia.' Previous work at the same station has estimated that DMS contributed to 73–79% and SO2 emissions from shipping activities ~21–27% of the non-sea-salt sulfate when taking into account data from all source area sectors (Li et al., 2018). We assume that the ratio would be similar for the bisulfate dimer. With the monomer the source regions were less clear since it is always the most abundant ion during the day.*

L352: What is the lifetime of DMS? Can it be transported to your site and get oxidized in the vicinity of your sampling site?

*The lifetime of DMS is approximately one day (see e.g., Kloster et al., 2006), so most of DMS would be lost on the way from the Antarctic coast, but we suppose some transport would be possible.*

Fig 8A: Could you insert more ticks to see the diameter range better? And please add number concentration of particles like you have in Fig 11.

*We have added more ticks and the PSM data as requested (see below). Originally we had left the PSM data out since the NAIS data already covers this size range relatively well.*

[Figure]

*Updated version of Figure 8.*

Fig 9, 10, 12,13: I would normalize the ion signals with TIC or just a use a diurnal plot of signals in order to see the increasing or decreasing signals. How many events in total (N) are depicted as NPF and non-events?

*We did originally also make versions where the signal was normalised, but felt that that was misleading since the TIC varies so clearly over the day. We made an example of Figure 9 with just the signals as lines (see reply to other reviewer), but we feel that the current way to present the data is easier to read.*

L406: The instrument measures naturally charged ions, so in that sense it is not the limitation of the instrument but actual phenomenon in the atmosphere. Bisulfates are produced during the day and (unfortunately) take the charge and hinder the identification of processes happening in the background.

*Yes this is true. We replaced 'likely due to instrumental limitation' by ' since we only measured naturally charged anions'.*

L440-450: How about correlation with formation rates? That could be better be suitable to particle numbers.

*Overall, only 28 formation rates were calculated and out of these, APi-TOF data was available only for 13 days, so we felt that we could not conclude much on such little data.*

Fig 11. I don't recommend using the size class 1-10 nm (if from NAIS), since the size classes 0.8-2nm (small ions) and 2-7 nm (intermediate ions) have very different dynamics and behaviour, see Manninen et al., 2016, fig 17. Therefore, I suggest you always separate them. The ones over 7 nm (large ions, 7-20 nm) should also be in different size class than the smaller ones. Make sure you depict the instruments used for the measurement in the figure caption. (Manninen et al., 2016, https://amt.copernicus.org/articles/9/3577/2016/amt-9-3577-2016.pdf)

*The 1-10 nm range is from PSM and SMPS, not NAIS. For part of the PSM data we only have this size range available. We added instrument information to the caption.*

L478: Did you have solar radiation measurements? You could show the link between the formation of precursors and radiation (or just use the daylight times)

*Global radiation data is available from the long term measurements of the station (see Peltola et al., 2022), but as mentioned earlier, we decided to leave meteorological data out of this manuscript. When we checked the data, the sum of bisulfate ions had a strong positive correlation with radiation (R = 0.71  in land-influenced air anr R = 0.78  in marine air), while MSA had a very weak positive correlation (R=0.057) with radiation in land-influenced air and weak negative correlation (-0.17) in marine air. We do not feel that this would bring new information to the manuscript.*

L482-485: I suggest you take the whole NPF period where small particles are formed, from 9-12 AM, and average it in one spectrum. Make a mass defect plot to interpret further, they are extremely useful tools for this type of analysis. I don't know what you want to say with the bisulfate signal decrease in the evening, since the formation event happens in the morning-noon?

*We added the sum mass spectrum and a quick mass defect plot with all fittable peaks from that time period below, but we would prefer not adding these to the manuscript. There are simply no peaks with masses higher than the bisulfate ion clustered with two sulfuric acid molecules, so we feel that the mass defect plot is not very useful in this particular case. With the evening decrease we refer to the period after 15 h, when the number concentration of 1-10 nm particles is elevated.*

[Figure]

*Mass spectrum from 9-12 h on October 15th 2020.*

[Figure]

*Mass defect plot from 9-12 h on October 15th 2020.*

L488: If you did the analysis based on the ion classes by Manninen et al., 2016, would this conclusion be different?

*Based on our earlier work and Figures B4-B6 we decided that ion data (at least in 2-4 nm size range) does not seem to be a good indicator of NPF in marine air and that is why we focused on the PSM data here.*

L495: I recommend you to look at mass defect plots and compare e.g. events to non-events, before and during event or marine to over land air masses.

*This is a good idea as such, but as the example mass defect just before hopefully shows, we believe that making these mass defect plots would most likely not give any more information than Figures 10 and 13 already do since not many compounds were identified (nor peaks fittable) apart from the compounds already in the figures.*

L501-505: Again, the ion size groups are selected a bit differently. 2-4 and <10 nm. I suggest using the standard operation procedure recommendations by Manninen et al., 2016 (0.8-2, 2-7 and 7-20 nm) if reporting NAIS data. This way the data will be further comparable with ACTRIS data also.

*We used 1-10 nm for PSM data, since this is all that was available for the whole PSM measurement period. From NAIS data 2-4 nm negative ions were used because this has been shown to be a good indicator of the initial steps of NPF (see e.g., Dada et al., 2018). As mentioned in our previous work (Peltola et al., 2022) we also discarded NAIS data above 15 nm due to instrumental issues, so using all the size classes from Manninen et al. (2016) would not be possible. Originally we did look also into sub-2 nm ions, but we left them out to keep the manuscript more concise.*

L523: Do you have values of what is the fraction of ion induced nucleation at your site? The APi-TOF should be capable of revealing clustering of iodate or bisulfates, you might need to average much longer times than 30 minutes to build up signal from NPF times and I suggest looking into mass defect plot or plotting spectra during event and non events on top of each other in order to find the differences.

*No, we did not try to estimate the fraction of ion induced nucleation and in marine air this would not even be possible since we did not have any traditional event for which we could calculate formation rates. As mentioned earlier, we did average the data over longer periods but no peak above ~400 amu were detectable.*

L548: concentration of neutral aerosol precursors? Total concentration of gases is a wide suggestion, but naturally recommended.

*Yes, we meant to refer more to the aerosol precursors. Both total and neutral concentrations could be measured if using an ion trap. We replaced 'total concentrations of gases and chemical clusters' by 'neutral or total concentrations of aerosol precursor species by'.*

Thank you for introducing the measurements to me and best of luck finishing the manuscript!

*Thank you!*

**Review comments: Chemical precursors of new particle formation in coastal New Zealand, M. Peltola et al., 2022**

This manuscript presents ion measurements from a remote site in the southern hemisphere using an atmospheric pressure interface time of flight spectrometer (APi-ToF) with no chemical ionisation inlet to capture ambient anions. This is an extensive dataset spanning seven months, and the accompaniment to a previous high quality paper published by the same authors (Peltola et al., 2022). The subject is of great scientific interest as it presents a comprehensive dataset in a highly unstudied area. The manuscript is well written, and figures are beautifully presented but would be greatly strengthened by some further analysis. Some key calculations can be done from the size distribution data (formation rates and ion-ion recombination rates at low diameters), some ion signals can possibly be assigned formulae with some careful thought, and if the averaging time is increased, some more mechanistic insight may be possible from the mass spectra. I highly recommend publication in ACP once a few comments are addressed.

*We appreciate the reviewer's comments and have answered them in more detail below, but here would like to just point out that we feel that the data set is somewhat limited since we do not have any peaks beyond ~400 Th no matter how long averaging times we use in the data analysis and thus we are not sure if identifying a few peaks more would really give us a more mechanistic understanding of the particle formation processes.*

Specific comments

Line 10: Should this say bisulphate, rather than sulphate? Or maybe sulphur-containing ions?

*Yes, bisulfate is more correct, we changed that.*

Line 69: If mentioning the sea surface microlayer, maybe reference (Mungall et al., 2017)

*We added that reference and changed sentence 'While the emissions of monoterpenes and isoprene are typically connected directly to biological activities (Shaw et al., 2010), isoprene can be produced also abiotically by photosensitized reactions at the sea surface microlayer (Ciuraru et al., 2015)' to 'While the emissions of monoterpenes and isoprene are typically connected directly to biological activities (Shaw et al., 2010), isoprene and oxidised VOCs can be produced also abiotically by photosensitized reactions at the sea surface microlayer (Ciuraru et al., 2015; Mungall et al., 2017)''*

Line 98: It would be nice to have some more details here about the running conditions for the PSM, I presume it was used in scanning mode, how long did each scan take, and how was the data inversion performed? Chan et al., 2020 presents four different techniques for inverting this data that give slightly different results. Was there any data pre-processing?

*The PSM was run first in fixed mode with supersaturation flow rate fixed at 1 lpm and then in stepping mode with saturation flow rate switching between 0.1 and 1 lpm every 60 s. As mentioned in the text, more details of the PSM measurements including this information can be found in Peltola et al. (2022). Since we did not use the scanning mode, we did not use any specific inversion code, we just assumed that a certain saturation flow rate corresponds approximately to a certain particle size (1 lpm to 1 nm and 0.1 lpm to 3 nm).*

Line 116: How many NPF events fall into this 4.3% of data coverage?

*As mentioned to the first reviewer, none of the traditional regional NPF events were observed in fully clean marine air.*

Line 117: How does this compare to the Potential Source Contribution Function? I.e., as discussed by (Fleming et al., 2012).

*The idea here is similar, but more simplified. One of the major differences is that we do not use a specific threshold for high concentrations like the PSCF but use an average of all concentrations.*

Line 130, Figure 2 & 3: These figures are very visually striking, but to me it is quite difficult to see the correlation between most of these species unless the correlations are particularly positive or negative. Would it make sense to reduce the alpha for lines corresponding to R values close to zero perhaps, as we are mostly concerned with stronger correlations?

*The idea here is to find the strongest correlations, not to be able to follow all the lines, since we had over 5000 correlations with p<=0.05. Changing the transparency of the line is not directly an option for this function and we are not sure if that would make a difference since the correlations close to zero are already coloured white and thus not visible. One option could be to set the lowest correlations (for example R<0.5) to zero. We made an example of this below, but did not add it to the manuscript as we feel that it is very similar to the figures we already have.*

[Figure]

*Figure: A different version of the schemaball plots with a corresponding to Figure 2 and b to Figure 3, but now with only correlations with coefficients above 0.5 are plotted.*

Line 139: "Pure sulfuric acid clusters were detected up to the trimer". Do you mean sulphuric acidbisulphate clusters? Also, is this referring to your dataset here, or Junninen et al.?

*Yes, we mean sulfuric acid - bisulfate clusters and we are refering to our work. We replaced the text 'Pure sulfuric acid clusters were detected' by 'We observed bisulfate-sulfuric acid clusters' to clarify this.*

Line 153: It would be nice to say how you went about deciding what peaks to assign formulae to, and what could not be fit. What possible combination of atoms did you look at? What error (i.e. in ppm) did you deem acceptable?

*We did not have a fixed ppm limit, but typically an error of around 10-20 ppm could have been acceptable if the isotopic pattern and peak composition also seemed reasonable. When searching for peaks we included different combinations of H, O, N, I, S, C, Br, Cl.*

*To clarify this process, we added text 'As a first approach we identified the most dominant peaks based on what has been seen in the literature before' to line 137. We also replaced 'we can observe' on line 142 by ' we looked for peaks containing H, O, N, I, S, C, Br, Cl, having an error below 10-20 ppm, fitting the isotopic pattern and having a reasonable chemical composition. This way, we could identify'.*

Line 171: Why is this a good reason to put them in another group? Do you infer that $HI_2O_6$ - has a different source than the other two iodine compounds from this?

*We do not think it has a different source, but we wanted to know if it would be an indicator of higher iodine oxoacid concentrations or an indicator of new particle formation from iodine oxoacids. For example bisulfate clustered with sulfuric acid has been observed to be a better indicator of particle formation than bisulfate anion alone and we wanted to see if we could see something similar with iodine oxoacids especially since we could not observe clusters at higher masses and follow the nucleation process further.*

Line 181: I'm not sure I understand the rationale here. Peaks in group "Other1" and "Other3" correlate with each other, but why is that important for the main thrust of the paper (aerosol formation?), especially when these contain 2 and 3 peaks, respectively. What counts as a "strong correlation" here? I understand the grouping of "Other2" and "Other4" as they correlate with aerosol concentrations (I presume this means total number concentrations from your CPC? Or is this N100 from integrating across the size distribution measurements). I will also echo the other reviewer here and suggest digging deeper to try and assign more of these mass spectral peaks. For example, your "Other4" group contains peaks which very well could be of the formulae $C_xH_yO_zN_1NO_3$ - . Some quick calculations show me that the peak at 339.024 m/Q would be about 150 ppm away from $C_{10}H_{15}O_8NNO_3$ - , and the peak at 373 about 150 ppm away from $C_{10}H_{17}O_{10}NNO_3$ - , the same with the two other ions. This discrepancy in mass could easily be mass calibration related perhaps

*It is true that these unidentified groups are not the most essential part of the paper. The idea behind these groups was that they showed promising correlations in the schemaballs and studying them further could bring us new knowledge either about particle composition or the composition of the ions. We were also hoping to point out that the list of identified compounds is not complete and we could be missing some interesting compounds. Considering a correlation strong required an R of at least over 0.5.*

*By aerosol concentrations we refer to different different number concentrations (N's), N1-10 (1-10 nm particles) uses both PSM and SMPS data, N10-100 (10-100 nm particles) and N100 (particles with diameter over 100 nm) use SMPS data. These have been defined in our previous work (Peltola et al., 2022), but we also added text '(1-10 nm based on PSM and SMPS data and 10-100 and >100 nm based on SMPS data)' to line 122 to clarify this.*

*For the peaks that you mention, we appreciate the suggestions of the chemical composition. They are reasonable  and it is very much possible that the mass calibration is off at higher masses since the signals of peaks at higher masses were so small (and/or unidentified) that we could not use them for mass calibration. We added a mention  of the potential chemical composition of the peaks in Group Other4 to line 186: 'For example peaks in group "Other4" are likely to follow formulae $C_xH_yO_zN_1NO_3$ -.' We would prefer not going back to modifying the peak list since running it for the whole data set would take a long time and would not necessarily bring that much new information to the paper.*

Line 194: It would be nice to substantiate the claim about halogen anions from the ocean with a reference

*We added a reference to Wang et al. (2021).*

Line 213: Are you confusing the NO3 radical with the nitrate anion here?

*Yes, we might be a bit confused here, we thought the diurnal cycles of the radical and the anion would be similar. We replaced the text 'This is because during the day, NO3 is destroyed by photolysis (e.g., Wayne et al., 1991).' by 'One possible daytime loss term of nitrate ions is the loss of charge to sulfuric acid. This has been seen before at other measurement sites (Eisele and Tanner, 1990; Yan et al., 2018).'*

Line 225: I'm not sure you can substantiate this claim with your data here, how do you know this is explained by a higher condensation sink, rather than a lower source strength? The CS has been calculated for this dataset (Peltola et al., 2022). It might be helpful to use this here.

*We added text 'both higher sources of SA in marine air and' before 'smaller losses'. We also added a comparison of the sulfuric acid condensation sinks and lifetimes in marine and land-influenced air after line 246 (see response to other reviewer).*

Line 235: Again, a reference r.e. these sources would be nice

*We added a reference to Wang et al. (2021).*

Line 256: As this discussion doesn't add much to the overall discussion of iodine oxides, why not include it with the sum of iodine anions?

*The idea in separating it was to see if we see different phenomena when the levels of iodine oxoacids are higher (see answer to comment for Line 171).*

Line 266: I am not sure organosulphate is the correct term for a HOM-bisulphate cluster as this usually refers to molecules with a R-SO4 - functional group

*Okay, we replaced 'organosulfates' with 'HOMs charged with bisulfate ions'.*

Line 282: This nighttime peak in Other4 is somewhat consistent again with what was observed for Organonitrate-nitrate anion clusters in previous work (Bianchi et al., 2017)

*Considering that all the compounds in this group also have odd masses, it is indeed possible that these are organonitrates with nitrate anion. However, in the work of Bianchi et al. (2017) the diurnal cycle has two maximums, one in the early morning and another in the evening whereas here we only observe one nighttime maximum. The chemical composition of these peaks could be studied more in the future.*

Figure 6: Why are groups Other1-4 not included here?

*We left them out to keep the figure more readable since even for diurnal cycles not much was seen for these groups. We did however check those plots and apart from higher wintertime land-influenced ion signals for group Other3, we did not see any clear seasonal cycles.*

Line 331: Just a note on this section: it may be useful also to try something like the potential source contribution function (PSCF) to simply identify the regions leading to the highest ion signals. This will exclude some of your data and perhaps highlight just the strongest source regions.

*Yes, this could be an option in the future, thank you.*

Line 340: The H2SO4HSO4 - cluster is a good indicator of NPF in Beijing, but what about this dataset?

*Yes, compared to the monomer or the sum of monomer, dimer and trimer, the trimer is typically a better indicator of NPF at Baring Head. See for example correlations with 2-4 nm ions in Figures B1 and B2 and correlation with 1-10 nm particles in Figure B5.*

Line 379: I think it should be possible to see some larger clusters during NPF events in the mass spectra, even with a suboptimal instrument tuning. Is it possible to average across the

entire NPF event? Possibly when the ion concentrations from the NAIS are elevated. This may help the data interpretation somewhat

*We averaged the data over several hours or even more than a day but were not able to see any peaks above ~400 amu.*

Figures 9, 10, 12, 13: It might be nice to see these in a non-stacked fashion to simply see how, for example, H2SO4HSO4 - behaves more clearly. These could simply go in the appendix.

*We tried to make Figure 9 non-stacked (see below), but feel that it is more difficult to read than the current plots since there are so many compounds with different levels of signal and that is why we would prefer sticking to the original plots.*

[Figure]

*This figure is the same as Figure 9 in the manuscript but using lines instead of stacked areas. Panel a is in linear scale and panel b in log scale.*

Line 427: Why not investigate the correlation with the formation rates? The J10 is available from the previous publication from this dataset, and the formation rate at lower sizes can be calculated from the available NAIS and PSM data, better yet, the ion-ion recombination rates can be calculated, which fit very nicely with the APi-ToF measurements. You may then find much better correlations with your ion signals.

*Overall, only 28 formation rates were calculated and out of these, APi-TOF data was available only for 13 days, so we felt that we could not conclude much on such little data. We are confused by what the reviewer means by ion-ion recombination rates fitting APi-TOF measurements.*

References Bianchi, F., Garmash, O., He, X., Yan, C., Iyer, S., Rosendahl, I., Xu, Z., Rissanen, M. P., Riva, M., Taipale, R., Sarnela, N., Petäjä, T., Worsnop, D. R., Kulmala, M., Ehn, M., and Junninen, H.: The role of highly oxygenated molecules (HOMs) in determining the composition of ambient ions in the boreal forest, Atmos. Chem. Phys., 17, 13819–13831, https://doi.org/10.5194/acp-17-13819-2017, 2017.

Chan, T., Cai, R., Ahonen, L. R., Liu, Y., Zhou, Y., Vanhanen, J., Dada, L., and Chao, Y.: Assessment of particle size magnifier inversion methods to obtain particle size distribution from atmospheric measurements, Atmos. Chem. Phys. Discuss., 1–21, 2020.

Fleming, Z. L., Monks, P. S., and Manning, A. J.: Review: Untangling the influence of air-mass history in interpreting observed atmospheric composition, Atmos. Res., 104–105, 1–39, https://doi.org/10.1016/j.atmosres.2011.09.009, 2012.

Mungall, E. L., Abbatt, J. P. D., Wentzell, J. J. B., Lee, A. K. Y., Thomas, J. L., Blais, M., Gosselin, M., Miller, L. A., Papakyriakou, T., Willis, M. D., and Liggio, J.: Microlayer source of oxygenated volatile organic compounds in the summertime marine Arctic boundary layer, Proc. Natl. Acad. Sci. U. S. A., 114, 6203–6208, https://doi.org/10.1073/pnas.1620571114, 2017.

Peltola, M., Rose, C., Trueblood, J. V., Gray, S., Harvey, M., and Sellegri, K.: New particle formation in coastal New Zealand with a focus on open-ocean air masses, Atmos. Chem. Phys., 22, 6231–6254, https://doi.org/10.5194/acp-22-6231-2022, 2022.

*References:*

*Beck, L. J., Sarnela, N., Junninen, H., Hoppe, C. J. M., Garmash, O., Bianchi, F., Riva, M., Rose, C., Peräkylä, O., Wimmer, D., Kausiala, O., Jokinen, T., Ahonen, L., Mikkilä, J., Hakala, J., He, X.-C., Kontkanen, J., Wolf, K. K. E., Cappelletti, D., Mazzola, M., Traversi, R., Petroselli, C., Viola, A. P., Vitale, V., Lange, R., Massling, A., Nøjgaard, J. K., Krejci, R., Karlsson, L., Zieger, P., Jang, S., Lee, K., Vakkari, V., Lampilahti, J., Thakur, R. C., Leino, K., Kangasluoma, J., Duplissy, E.-M., Siivola, E., Marbouti, M., Tham, Y. J., Saiz-Lopez, A., Petäjä, T., Ehn, M., Worsnop, D. R., Skov, H., Kulmala, M., Kerminen, V.-M., and Sipilä, M.: Differing mechanisms of new particle formation at two Arctic sites., Geophysical Research Letters, 48, e2020GL091 334, 2021*

*Dada, L., Chellapermal, R., Buenrostro Mazon, S., Paasonen, P., Lampilahti, J., Manninen, H. E., Junninen, H., Petäjä, T., Kerminen, V.-M., and Kulmala, M.: Refined classification and characterization of atmospheric new-particle formation events using air ions, Atmospheric Chemistry and Physics, 18, 17 883–17 893, 2018*

*Ehn, M., Thornton, J. A., Kleist, E., Sipilä, M., Junninen, H., Pullinen, I., Springer, M., Rubach, F., Tillmann, R., Lee, B., et al.: A large source of low-volatility secondary organic aerosol, Nature, 506, 476, 2014.*

*Kloster, S., Feichter, J., Maier-Reimer, E., Six, K. D., Stier, P., and Wetzel, P.: DMS cycle in the marine ocean-atmosphere system - a global model study, Biogeosciences, 3, 29–51, https://doi.org/10.5194/bg-3-29-2006, https://bg.copernicus.org/articles/3/29/2006/, 2006.*

*Li, J., Michalski, G., Davy, P., Harvey, M., Katzman, T., and Wilkins, B.: Investigating Source Contributions of Size-Aggregated Aerosols Collected in Southern Ocean and Baring Head, New Zealand Using Sulfur Isotopes, Geophysical Research Letters, 45, 3717–3727, 2018.*

*Wang, X., Jacob, D. J., Downs, W., Zhai, S., Zhu, L., Shah, V., Holmes, C. D., Sherwen, T., Alexander, B., Evans, M. J., et al.: Global tropospheric halogen (Cl, Br, I) chemistry and its impact on oxidants, Atmospheric Chemistry and Physics, 21, 13 973–13 996, 2021*

---

## Author Response (AR2)

Dear Authors:

Thank you for your response to the referee comments and your manuscript revisions. Both referees expressed that the manuscript would be strengthened by further calculations/analysis and one felt that it should be recharacterized as a measurement report if no further analysis was performed. Given the unique aspects of the dataset and the analysis included, I support publication as a research article. After carefully considering all the documents, it is my opinion that several revisions are required before the manuscript is suitable for publication in ACP. Several of these relate to instances where responses to referee comments were informative in the response document but were not included in edits to the main document. Please consider the comments below. Unless otherwise noted, line numbers refer to the track changes version of the manuscript.

*Dear editor,*

*Thank you for your helpful comments. We have addressed them below in grey italics.*

Main comments

1. I suggest you consider rewriting many of the chemical formulas to better reflect the ions. For some (non-exhaustive) examples $HI_2O_6-$ is probably better written as $(HIO_3)IO_3-$, $(SO_4)H_3O-$ is probably better as $(H_2O)HSO_4-$, $(SO_4)_3H_5-$ is better as $(H_2SO_4)_2HSO_4-$, etc. This will be more aligned with how these ions have been represented in past works and also increases the ease of reading.
*We have gone through the document and tried to improve this.*
2. Section 2.1: Please add more details regarding the PSM even though more extensive details are given in the cited work. It is important that the basics are conveyed without having to reference another manuscript. Something along the response given to Dr. Brean's comment regarding line 98 (of the original document) is sufficient.
*We added the text: "The PSM was first used in fixed mode with a saturation flow rate of 1 lpm and from 17 September it was used in fixing mode, with the saturation flow switching from 0.1 to 1 lpm every 60 s. Saturation flow rate of 1 lpm was assumed to correspond to approximately a cutoff diameter 1 nm and 0.1 lpm to 3 nm."*
3. Line 117: Please add a sentence or two similar to the response given to Dr. Brean's comment regarding line 117 (of the original document) regarding the similarities/differences to PSCF.
*We added the text: "This method is similar to the Potential Source Contribution Function (PSCF) method described by Fleming et al. (2012) , but more simplified. One of the major differences is that we do not use a specific threshold for high concentrations like the PSCF but use an average of all concentrations."*
4. Line 144-145: "…$(NH_3(H_2SO_4)_3HSO_4-$, in the peaklist unknown0085)." is confusing wording to the reader. I suggest that you make it clear that this peak was assigned later on and provide the ppm difference between the assignment and the unknown. Please also include the assignment in the peak list.

*We have replaced the text "in the peaklist unknown0085" by "please note that this peak was only identified after processing the data and that is why the mass of the peak is 13 ppm off" and changed the name in the peaklist from unknown0085 to the expected composition.*

5. Lines 153-155: The wording here makes it sound like the lack of peaks can be quantitatively explained by these factors. I suggest rewording to make it clear these are possible reasons.

*We replaced the text "The lack of peaks at higher masses can be explained by a combination of" by "Possible reasons for the lack of peaks at higher masses include".*

6. Figure 1: I suggest considering enhancing the intensity of the peaks above approximately m/z 250 by using a scaling factor. Alternatively, a log scale y-axis may make it easier to see some of the peaks.

*We thought about this and ended up adding a second panel that shows the masses between 250-500 m/Q better (see below) Masses above 500 m/Q were not included, since there were no peaks and this way the peaks in this range are more clear.)*

[Figure]

7. Figures 2 & 3: Personally, I found the examples presented in the response to questions posed by both referees more compelling as it further enhanced the strong relationships that are discussed in the manuscript. Particularly the strong positive

correlations get lost when overlaid on the numerous slightly positive correlations. Additionally, I'm not sure if this is possible, but is there a way (perhaps using coloring) to identify the ions that are called out in table 2? That would help the reader follow the arguments made in the manuscript.

*We have replaced the figures with versions that only contain lines for correlation coefficients above 0.5 and bolded the labels for the compounds that were picked for the list (see below).*

[Figure]

*Updated version of Figure 2.*

[Figure]

*Updated version of Figure 3.*

8. Line 295: Following up on Dr. Brean's comment, the diel profile may be more consistent with organonitrate production from NO3 radical chemistry than from OH + NOx chemistry. This could plausibly explain the difference in the diel profile between your work and the Bianchi et al work.

*Yes, this sounds possible. We added text ", potentially organitrates produced through NO3 radical chemistry and clustered with a nitrate anion (see e.g., Yan et al, 2016)", to the end of the paragraph.*

9. Line 355: I suggest including the further information on the attribution of non-sea-salt sulfate from DMS and SO2 from Australia provided in response to Dr. Jokinen's comment on line 344.

*We added text "The previous work by Li et al. (2018) estimated that DMS contributed to 73–79% and SO2 emissions from shipping activities ~21–27% of the non-sea-salt sulfate when taking into account data from all source area sectors. We assume that the ratio would be similar for the bisulfate dimer."*

10. Figure 8: I don't understand why from ~12:00-19:00, the N1-10 timeseries in the lower panel of (a) is at zero while the upper panel shows high counts in the 2-10 nm region for most of the time period and N1-3 is also non-zero. Is it because these are all from different instruments? If so, how does this disagreement impact conclusions/interpretations about particle counts at these lowest sizes throughout the manuscript?

*Yes, as stated in the caption, these are all from different instruments that have different inlets and function differently. The N2-10 range of the upper panel is using NAIS data whereas N1-10 is calculated using both PSM and SMPS data and N1-3 using only PSM data. Both PSM and SMPS count particles with CPC's that grow the particles by condensing butanol on them and then count them optically. The NAIS on the other hand uses unipolar chargers, differential mobility analyzers and electrometers to determine the particle size distribution. In the NAIS, the losses of small particles are clearly lower since the NAIS has a higher flow rate (60 lpm) and larger inlet (diameter 2.5 cm) tubing whereas the inlets for PSM and SMPS were only ¼" wide and had an order of magnitude lower flow rates.*

*Since N1-10 uses two different instruments which can have different sensibilities it is more uncertain and it is possible for the value to go to zero. On the other hand N1-3 also has its own uncertainties since this is a rather narrow size range and the PSM is sensitive to environmental conditions and the losses of these small particles are higher than those of larger particles.*

*In Section 3.5 where we connect aerosol data to APi-TOF data we focus on SMPS data and 2-4 nm anions from NAIS data for land-influenced NPF and PSM and SMPS data for marine NPF and we do not really use the neutral NAIS data to interpret the APi-TOF data.*

*In the beginning of Section 3.5.2 we mention 'Our previous work showed that in clean marine air, new particle formation should not be studied with the traditional criteria used for continental sites. Instead we focus on particle growth episodes and appearance of sub-10 nm particles. -- We also show PSM data for sub-10 nm particles instead of NAIS data, since NAIS seemed to underestimate particle concentrations which is especially problematic in the clean marine air where the concentrations are low.' We have now*

*added text ' Here it should be noted that the NAIS and PSM have different functioning principles and hence also different uncertainties. In the PSM data,.' after this to alert the reader of the uncertainties in the data. Throughout the text there are also several mentions of when the 2-4 nm anions and N1-10 are in agreement and when not and we hope that the reader can make their own conclusions on this.*

11. Discussion starting at line 423: API-TOF is best suited for capturing NPF happening at the measurement location and thus events identified as transported events in your previous work would likely not be observable as the characteristic NPF API-TOF peaks. I think it would strengthen the manuscript to include some of the information provided in your previous manuscript about if any/all of these were classified as transported events by your previous analysis and thus might explain the lack of larger ions observed by the API-TOF. Are there differences in the ions observed between transported and regional event days?

*This is an interesting point. The figure that compares event and non-event days considers event days to be Class I and Class II events. Our previous work showed that out of Class I 30% were transported events whereas out of Class II events only 10.5% were classified as transported events (when data from both classifications was available). In total that makes 20.5% of the events considered here, so presumably the figure should be dominated by regional events. We did a quick check to see if the levels of different compounds (here just HSO4-, CH3O3S-, IO3-, C10H14O9NO3-) differ on regional and transported event days, but did not see any statistically significant differences, so we expect that the transported events do not significantly change the results.*

12. Figures 9 and 10: I agree with both referees that it would be nice to see this version in a non-stacked view so that the reader can more clearly see the diel variations. I agree with your point in the response that the figure is too busy when all of the ions are included. However, many of the ions/ion groups are extremely low and poorly resolved in the current figure as well. I think a simplified figure including the most important ions (either because of overall intensity or because they are discussed substantially in the manuscript) would be appropriate. Placing such a figure in supporting information would be appropriate. In Figure 10, please state the number of event and non-event days.

*We have made simplified plots for Figures 9 and 10 and for consistency also figures 12 and 13 and added them to the supplement (see below). There were 53 event days and 90 non-event days with APi-TOF data available and we have added this information to the caption of Figure 10.*

[Figure]

*Additional figure for Figure 9.*

[Figure]

*Additional figure for Figure 10.*

[Figure]

*Additional figure for Figure 12.*

[Figure]

**(a) N1-10 > 500 cm-3**   **(b) N1-10 < 500 cm-3**

*Additional figure for Figure 13.*

Technical

1. Line 106: Please specify the inner and outer diameters of the tubing. Please also provide the flow rate into the instrument.

*We have replaced "3/8" thick stainless steel inlet" by "stainless steel inlet with an outer diameter of 3/8" and thickness of approximately 1 mm" and added text "The instrument flow rate was 0.8 lpm, but" to the beginning of the next sentence.*

2. Both m/Z and m/Q are used in the manuscript. Please standardize.

*Thank you for pointing this out. We have changed all m/Z to m/Q.*

3. Lines 169-170: Everything in the list with the exception of HOMs are given in their ion form. I suggest saying identifying HOMs as the observed ion (e.g., HOMs clustered with NO3-).

*Done.*

4. Lines 257-258: Please include units on the condensation sink values.

*Done.*

5. Figure 6: Please include a legend for land and marine. I also find this figure very difficult to read as the panels are small and the seasonal variations are muted due to the use of a log scale. In figure 5, the diel cycles were more apparent making the figure easier to interpret than this one. I suggest considering only showing the most important panels in Figure 6 in the main text and making them larger. The other ones can be put into supporting information. Would a non log plot help make the seasonal cycles more apparent?

*We have added the legend, divided the figure into two figures and replaced text 'We use only 10 groups' by 'Here we show six of the most important compounds (Fig. 6) and in the supplementary four more compounds (Fig. B2).'*

[Figure]

*Figure to replace Figure 6.*

[Figure]

*Figure B2 that will complement Figure 6.*

6. Section 3.5.2: I think the name of this section should be modified to reflect that it discusses sub 10 nm particle days and not specifically true NPF.

*In addition to sub-10 nm particles, this section also discusses later stage growth and even if we do not see traditional banana type events, which only occur in homogeneous air masses, we consider observing sub-10 nm particles as NPF.*

*References:*

*Yan, C., Nie, W., Äijälä, M., Rissanen, M. P., Canagaratna, M. R., Massoli, P., Junninen, H., Jokinen, T., Sarnela, N., Häme, S. A., et al.: Source characterization of highly oxidized multifunctional compounds in a boreal forest environment using positive matrix factorization, Atmospheric Chemistry and Physics, 16, 12 715–12 731, 2016.*